# The Evolution of Internal Damage Identified by Means of X-ray Computed Tomography in Two Steels and the Ensuing Relation with Gurson's Numerical Modelling

**Fernando Suárez** [1] , **Federico Sket** [2] , **Jaime C. Gálvez** [3,*] , **David A. Cendón** [4] , **José M. Atienza** [4] and **Jon Molina-Aldareguia** [2]

1   Departamento de Ingeniería Mecánica y Minera, Universidad de Jaén, 23071 Jaén, Spain; fsuarez@ujaen.es
2   Instituto IMDEA Materiales, C/Eric Kandel 2, Tecnogetafe, 28906 Madrid, Spain; federico.sket@imdea.org (F.S.); jon.molina@imdea.org (J.M.-A.)
3   Departamento de Ingeniería Civil-Construcción, Universidad Politécnica de Madrid, E.T.S.I. Caminos, Canales y Puertos, 28040 Madrid, Spain
4   Departamento de Ciencia de Materiales, Universidad Politécnica de Madrid, E.T.S.I. Caminos, Canales y Puertos, 28040 Madrid, Spain; david.cendon.franco@upm.es (D.A.C.); josemiguel.atienza@upm.es (J.M.A.)
*   Correspondence: jaime.galvez@upm.es; Tel.: +34-913-365-350

**Abstract:** This paper analyzes the evolution of the internal damage in two types of steel that show different fracture behaviors, with one of them being the initial material used for manufacturing prestressing steel wires, and the other one being a standard steel used in reinforced concrete structures. The first of them shows a flat fracture surface perpendicular to the loading direction while the second one shows the typical cup-cone surface. 3 mm-diameter cylindrical specimens are tested with a tensile test carried out in several loading steps and, after each of them, unloaded and analyzed with X-ray tomography, which allows detection of internal damage throughout the tensile test. In the steel used for reinforcement, damage is developed progressively in the whole specimen, as predicted by Gurson-type models, while in the steel used for manufacturing prestressing steel-wire, damage is developed only in the very last part of the test. In addition to the experimental study, a numerical analysis is carried out by means of the finite element method by using a Gurson model to reproduce the material behavior.

**Keywords:** steel; tensile test; XRCT; damage evolution; Gurson model

## 1. Introduction

Steel is, with concrete, the most important building material used currently, and the standard tensile test is the most widely used method to determine its principal material properties [1]. With it, the stress–strain curve of the material may be obtained before the ultimate tensile strength is reached; stresses and strains are then difficult to estimate. Therefore, the material behavior after the ultimate tensile strength is usually neglected. Nevertheless, understanding the fracture mechanisms that lead to the eventual failure is of significant interest, since it can help to improve structural safety strategies.

The fracture of ductile materials is usually explained with the theory of nucleation, growth, and coalescence of microvoids [2]. According to this theory, at a first stage and with high loading being applied, microvoids develop inside the material (nucleation) as a result of the decohesion of small inclusions that are torn apart from the rest of the material or by the fracture of these inclusions.

At a second stage, and under higher strain states, these nucleated voids increase their size (growth) until, at the last stage, they become interconnected (coalescence). This process weakens the material leading to its eventual failure.

This theory has given rise to many mathematical models, with a special mention to the Gurson model [3]. This model was proposed in 1977 and its formulation was based on how a spherical void grows inside a material matrix under certain stress states. Subsequent evolutions of the model have been able to reproduce not only softening due to the growth of microvoids, but also the eventual fracture of the material [4]. Since its appearance, the models based on Gurson's formulation, that here will be referred to as Gurson-type models, have been applied to reproduce damage evolution in different ductile materials, such as aluminum, copper and steel [5–7]. When such models are used to reproduce a tensile test on a ductile material, the damage begins to develop in very early stages of the test, as shown in Figure 1, even before the onset of necking, which takes place approximately when the maximum load is reached.

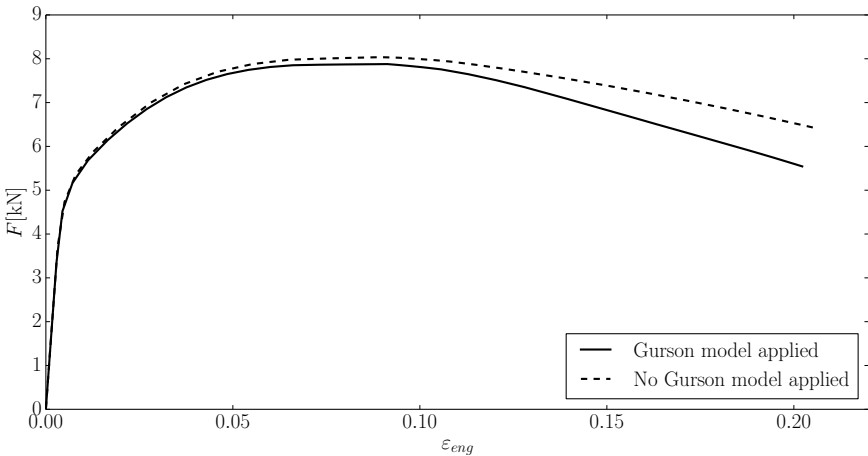

**Figure 1.** F-$\varepsilon$ curves obtained with numerical simulations of the same specimen. One of them using a Gurson model and the other one without it.

Steel properties may differ depending on many factors, such as the chemical composition and the manufacturing process used. Such differences may affect ductility and the fracture mechanisms that lead to failure, with geometrically identical cylindrical specimens of different steels presenting a contrasting fracture surface after a tensile test. Figure 2 shows the fracture surface of two 9 mm-diameter specimens made of different steels. The one on the right shows a typical cup-cone fracture, usually observed on ductile materials [2] and which belongs to a specimen made of standard steel used in reinforced concrete structures. The one on the left shows a flat fracture surface, where a dark circular region can be observed, belonging to a specimen made of steel used for manufacturing prestressing steel wires.

In this work, two steels that have distinct fracture behaviors are analyzed. The first of them corresponds to the initial material used to manufacture prestressing steel wires (Material 1) and exhibits the flat fracture pattern shown in Figure 2a. The second material corresponds to a standard steel used in reinforced concrete structures (Material 2) and presents a fracture surface corresponding to the classical cup-cone shape (Figure 2b). These materials have been analyzed in the past and their mechanical properties are well known; for further information, the reader is referred to [8–10]. In the first place, the fracture surfaces of both materials are observed with a scanning electronic microscope to identify the fracture mechanisms that take place. In the second place, an analysis of damage evolution along a tensile test on cylindrical steel specimens is carried out; this analysis is performed for both materials. Specimens are tested in subsequent loading steps and, at the end of each step, analyzed with X-ray computed tomography (XRCT) to identify the evolution of the

internal damage. Maire et al. [11–13] have used a similar approach on aluminum and steel specimens to quantify damage evolution and study the effect of distinct triaxiality states on this process, which has served to compare experimental values of void growth and damage evolution with those predicted by numerical models [14]. It must be noted that in this work the evolution of internal porosity is analyzed, but also the appearance of true cracks inside the specimen before the eventual failure. For this reason, the term damage is used as a general concept that refers to voids, which may nucleate and grow, and cracks, which can be the result of coalescence of voids, both processes weakening the material.

(**a**)　　　　　　　　　　　　　　　　　　　　　　　　　　(**b**)

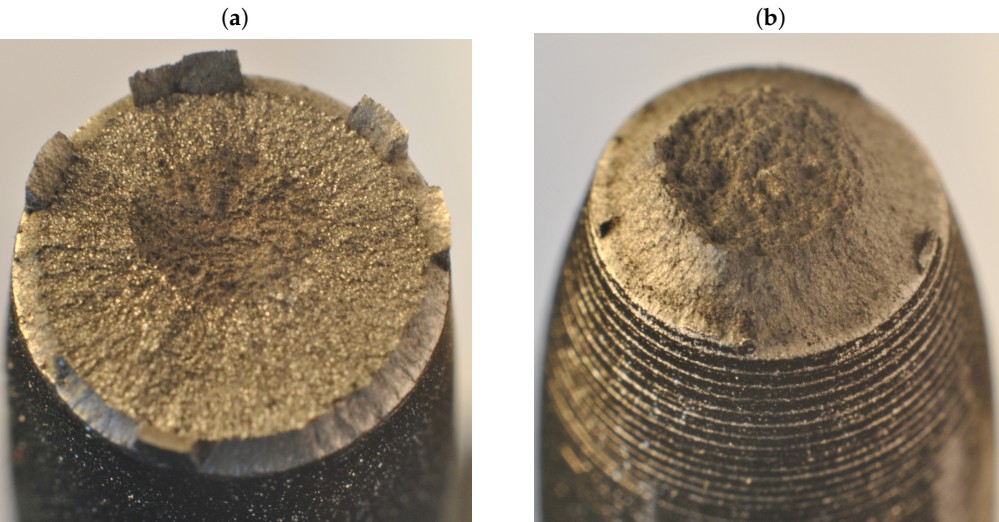

**Figure 2.** Fracture surfaces on 9mm-diameter specimens of two steels with different fracture patterns after testing under tension: (**a**) Material 1; (**b**) Material 2 [8].

To study how one of the most extended numerical models used with metals, the Gurson model, reproduces damage evolution, the tensile test for both materials is numerically reproduced. This numerical study is carried out using the finite element method and the material behavior is reproduced by the Gurson model. Although this model was first proposed in 1977 [3], it is extensively used presently with variations in many fields such as metal forming [15,16]. There exist several variations that allow its formulation to adapt for different materials or load cases [7,17–21]. Here, the model is calibrated by using macroscopic results obtained experimentally: the load-strain curve and the necking radius evolution. The evolution of the internal porosity numerically predicted is compared with the experimental values provided by XRCT.

## 2. Experimental Work

### 2.1. Materials

Two different steels are considered in this work. This section describes their characteristics.

### 2.1.1. Material 1

To manufacture steel wires, raw eutectoid steel bars are driven through a cold-drawing process that reduces their section by pulling them through a conical die. This process affects the failure behavior of the material [22,23] and may introduce additional uncertainties in this study. For this research, Material 1 specimens were obtained from raw eutectoid pearlitic steel bars used to produce prestressing steel wires, thus not being affected by cold-drawing, which provided a material that was as isotropic as possible. The chemical composition of this material can be consulted in Table 1. According to the metallographic analysis [24], the microstructure of this material is formed by perlite, with a lamellar structure of equi-axed grains with an average size of *G* = 9. This microstructure can be seen in Figure 3a.

**Table 1.** Chemical composition of both materials in %.

| Material | C | Si | Mn | P | S | Cr | Mo |
|---|---|---|---|---|---|---|---|
| **1** | 0.83 | 0.25 | 0.72 | 0.012 | 0.004 | 0.24 | <0.01 |
| **2** | 0.22 | 0.18 | 1.00 | 0.024 | 0.042 | 0.08 | 0.03 |
| **Material** | Ni | Cu | Al | Ti | Nb | V | N |
| **1** | 0.02 | 0.01 | <0.003 | <0.005 | <0.005 | <0.01 | 0.0097 |
| **2** | 0.14 | 0.46 | <0.003 | <0.005 | <0.005 | <0.01 | 0.0113 |

### 2.1.2. Material 2

Specimens of Material 2 are made of steel reinforcement B 500 C, according to the European Standard EN 10020 [25], with high ductility and an elastic limit of 500 N/mm². The chemical composition of this material can be consulted in Table 1. According to the aforementioned metallographic analysis [24], the microstructure of this material is formed by ferrite and perlite in a proportion of 50/50, with the perlite being sorbitic and equi-axed grains with an average size of $G = 9$. This microstructure can be seen in Figure 3b.

(**a**)  (**b**)

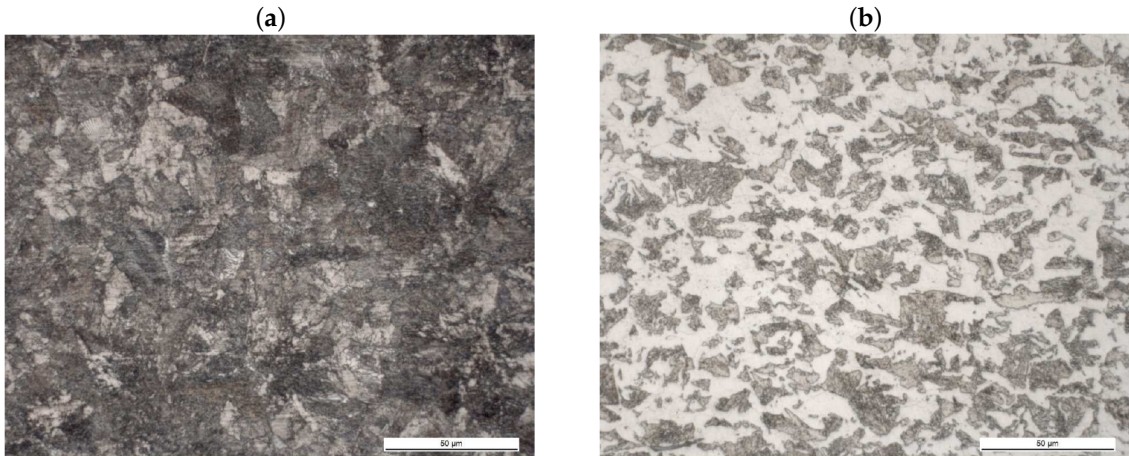

**Figure 3.** Microstructure of both materials in the longitudinal direction before testing. (**a**) Material 1 ($\varepsilon_{eng} \approx 0$). (**b**) Material 2 ($\varepsilon_{eng} \approx 0$).

### 2.2. Specimens

Two diameters were considered for each material: 3 mm and 9 mm. The 9 mm-diameter specimens were used to analyze the fracture surfaces, since comparison is easier in specimens with larger cross sections. The 3 mm-diameter specimens were used in the damage evolution analysis to ensure a correct penetration of the X-rays used to identify the internal damage. The dimensions of these specimens can be checked in Figure 4.

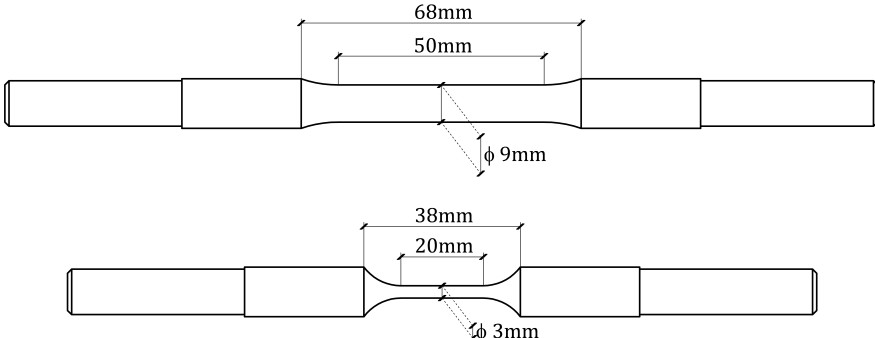

**Figure 4.** Specimens dimensions.

## 2.3. Testing Procedure Used to Study the Damage Evolution

To follow the damage evolution inside a specimen, it was tested in subsequent load steps, with the first one being the strain value at which the maximum load is reached. Once each strain value was reached, the specimen was unloaded and its neck analyzed with XRCT (Nanotom 160NF, Phoenix, General Electric, Wunstrof, Germany).

The process for each analysis is described as follows:

1.  X-ray tomographic analysis before the specimen is tested.
2.  The specimen is tested until the maximum load is reached, which is identified as Step 1; then, it is unloaded.
3.  X-ray tomographic analysis of the specimen.
4.  The specimen is tested until the second step is reached; then, it is unloaded.
5.  X-ray tomographic analysis of the specimen.
6.  The previous actions are repeated for subsequent steps until the point of failure.

The tensile tests were carried out with a Suzpecar universal testing machine and a load cell of 100 kN with an accuracy of 0.5%. Every load step was performed with a displacement control, and at a very low speed of displacement of the clamping jaws (of the order of 0.05 mm/min in the last steps), to avoid dynamic effects during the test. The tomographic images were collected at 80 kV and 140 μA by using a tungsten target.

## 2.4. Results

### 2.4.1. Fractographic Analysis of the Fracture Surfaces

Figures 5 and 6 show the fractographs obtained with the 9 mm-diameter specimens tested, with the former corresponding to the Material 1 specimen and the latter to the Material 2 specimen. In the Material 1 specimen, the detail of the central region corresponds to the internal dark region observed after a tensile test and the external region refers to the lighter area that surrounds it. In the Material 2 specimen, the detail of the central region corresponds to the flat surface perpendicular to the specimen axis in a cup-cone fracture surface and the external region refers to the inclined lips that develop around it.

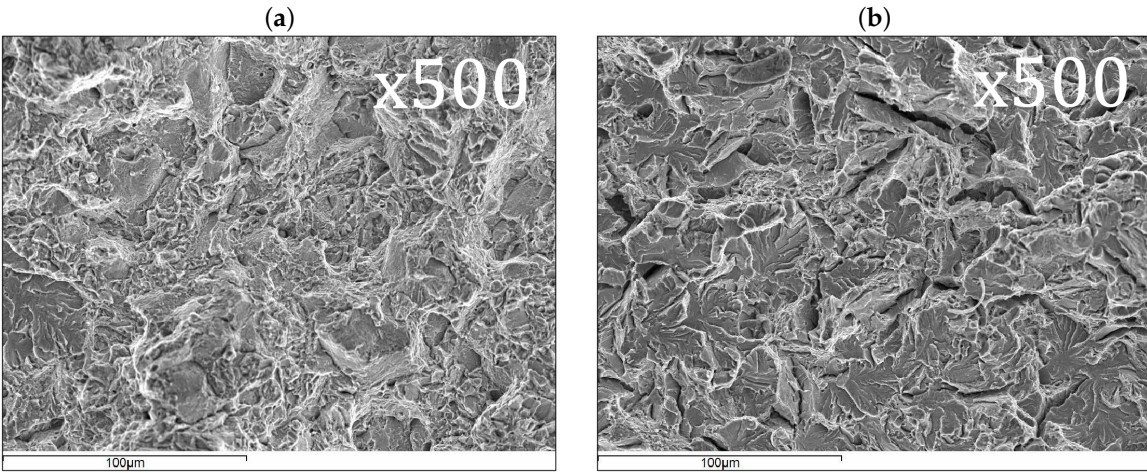

**Figure 5.** Fractographs obtained from the 9 mm-diameter specimen made of Material 1 [26]. (**a**) Detail of the central region. (**b**) Detail of the external region.

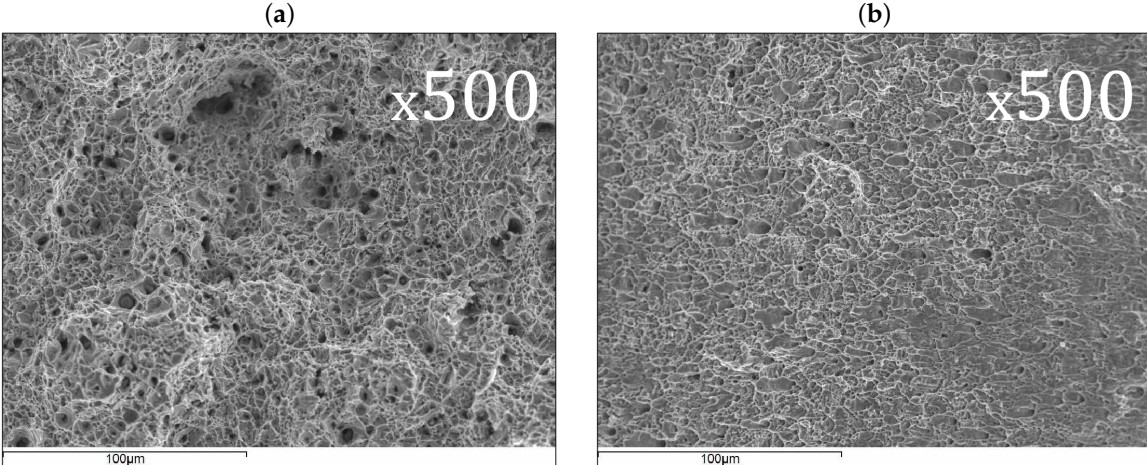

**Figure 6.** Fractographs obtained from the 9 mm-diameter specimen made of Material 2 [26]. (**a**) Detail of the central region (cup). (**b**) Detail of the external region (cone).

The fractographic pictures of the Material 1 specimen show that the central region presents a surface with dimples, different from that of the external region, where sharper edges are noticed. According to this, the central region fails by means of a nucleation, growth, and coalescence mechanism, while the external region corresponds to a cleavage fracture mechanism. In the case of the Material 2 specimen, the central region presents a high number of voids and evident dimples, which corresponds to a clear nucleation, growth, and coalescence mechanism. The external region presents fewer voids and slightly sharp edges, which seems to indicate a combination of the nucleation, growth, and coalescence mechanism with the cleavage mechanism.

2.4.2. Metallographic Analysis after the Test

Figure 7 shows the microstructure in the longitudinal direction of Material 1 and Material 2 after the test in the necking region.

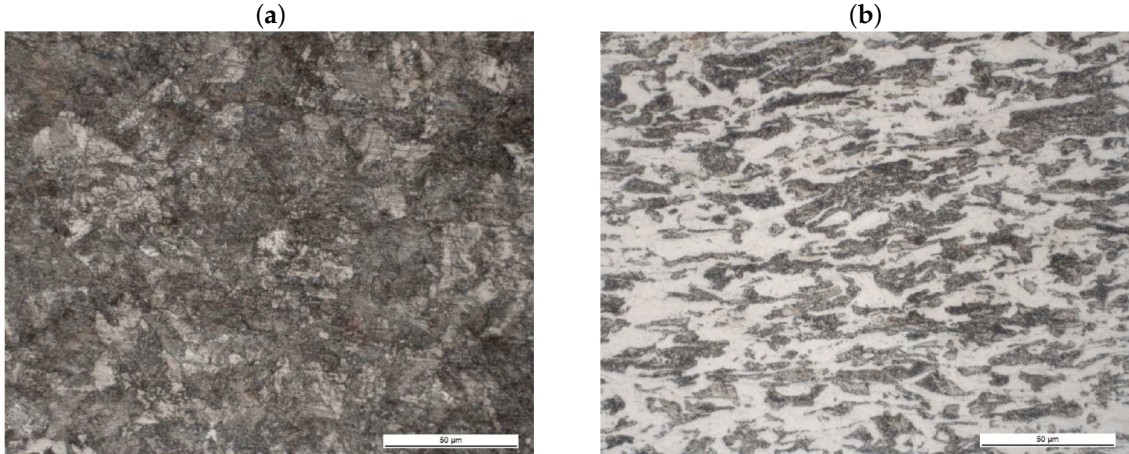

**Figure 7.** Microstructure of both materials in the longitudinal direction after testing. (**a**) Material 1 ($\varepsilon_{eng} \approx 0.15$). (**b**) Material 2 ($\varepsilon_{eng} \approx 0.32$).

In the case of Material 1, raw eutectoid pearlitic steel, no differences can be observed when compared with the microstructure before testing (see Figure 3a). In the case of Material 2, standard steel used as reinforcement in concrete structures, grains are oriented in the longitudinal direction after the test and show a shape factor of $f_{sh} = 4$.

### 2.4.3. Internal Damage Evolution Analysis

This analysis was performed with 3 mm-diameter specimens to allow penetration of X-rays into the material. Please note that some of the results presented here were already included in [26], but are now completed with more figures that help to better describe the damage evolution process in each material.

Since XRCT is an expensive technique that requires much postprocessing work, only one specimen for each material is analyzed, which is usual practice when this technique is applied [27–29]. The analysis was carried out with a voxel size of 2.5 µm (Nanotom 160NF, from Phoenix X-ray). As mentioned before, to carry out this analysis the tensile tests were performed in subsequent load steps. These steps are defined by the engineering strain developed along an initial length of 12.5 mm and can be consulted in Table 2. Figure 8 shows them over the corresponding F-$\varepsilon_{eng}$ curves. Please note that this figure shows the experimental results for both analyzed materials compared with the numerical results that will be described later.

**Table 2.** Steps used for the damage evolution analysis with the specimens of both materials. Engineering strain over a 12.5 mm initial length is used for identifying every step.

| Step | 1 | 2 | 3 | 4 | 5 |
|---|---|---|---|---|---|
| **Material 1** | 0.076 | 0.104 | 0.118 | 0.128 | — |
| **Material 2** | 0.193 | 0.267 | 0.288 | 0.306 | 0.319 |

Figure 9 presents the results for the Material 1 specimen. Results are given for the specimen before testing and for each of the four steps considered. Three pictures are shown for each step: a longitudinal section of the necking region, a projection of damage on the cross section and a perspective of the internal damage in the necking region.

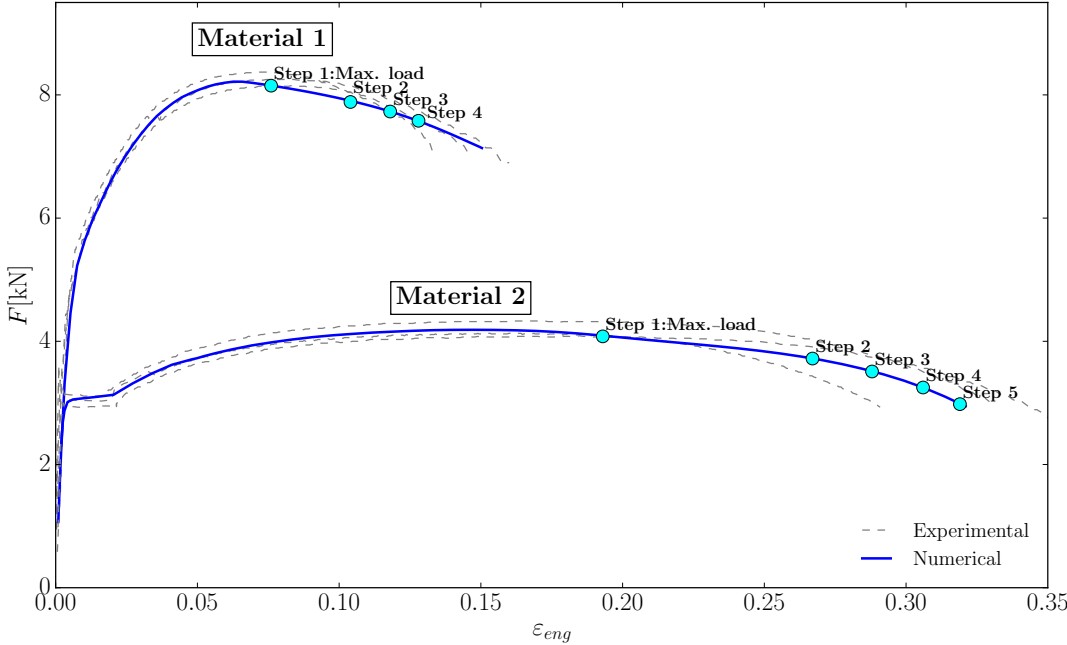

**Figure 8.** Load-strain curve of the 3 mm-diameter specimen of both materials; comparison between the experimental results and the numerical models. The steps used for the damage evolution analysis are represented with blue circles (the engineering strain has been obtained using an initial gauge length of 12.5 mm).

According to these results, nucleation and growth of microvoids is unnoticeable in the first step, which corresponds to the maximum load instant. In the second and third steps, the mechanism of nucleation and growth of microvoids can be clearly observed, though it is not until the fourth step that an evident internal damage is developed. Therefore, according to the theory of nucleation, growth, and coalescence of microvoids, as the test progresses the microvoids start to appear and grow in an even manner but the internal damage that leads the specimen to failure only appears at the very end of the test.

Figure 10 shows the results for the Material 2 specimen. As in the Material 1 specimen, results are given for before testing and for each of the steps considered, five in this case. As in Figure 9, three pictures are shown for each step: a longitudinal section of the necking region, a projection of damage on the floor plan and a perspective of the internal damage in the necking region.

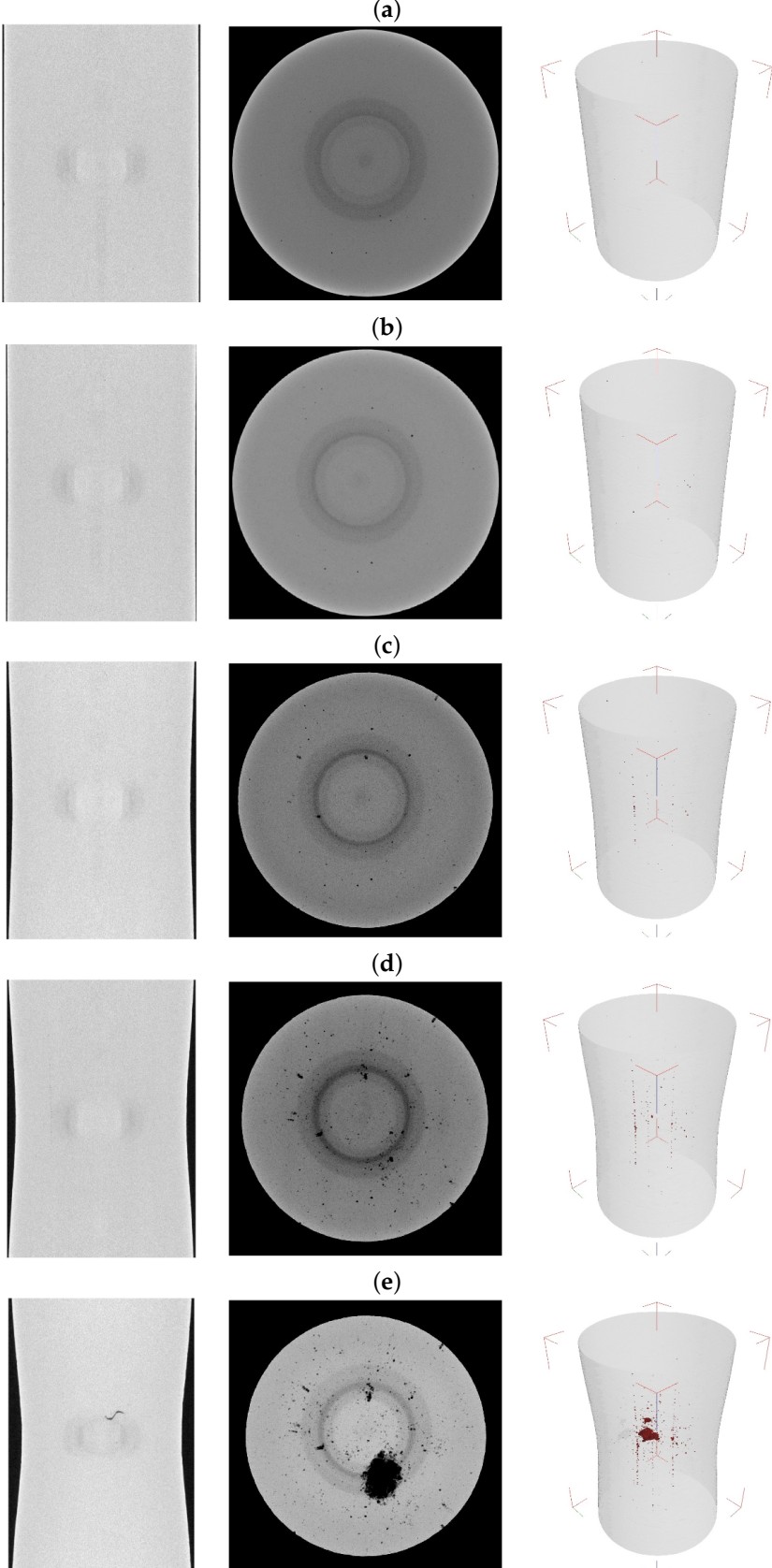

**Figure 9.** Results of the XRCT analysis of the necking zone for Material 1. For each step, three results are given: (I) Longitudinal section, (II) Damage projected on the floor plane and (III) Perspective of the internal damage. (**a**) Step 0: Before testing ($\varepsilon_{eng} = 0$). (**b**) Step 1: Maximum load ($\varepsilon_{eng} = 0.076$). (**c**) Step 2 ($\varepsilon_{eng} = 0.104$). (**d**) Step 3 ($\varepsilon_{eng} = 0.118$). (**e**) Step 4 ($\varepsilon_{eng} = 0.128$).

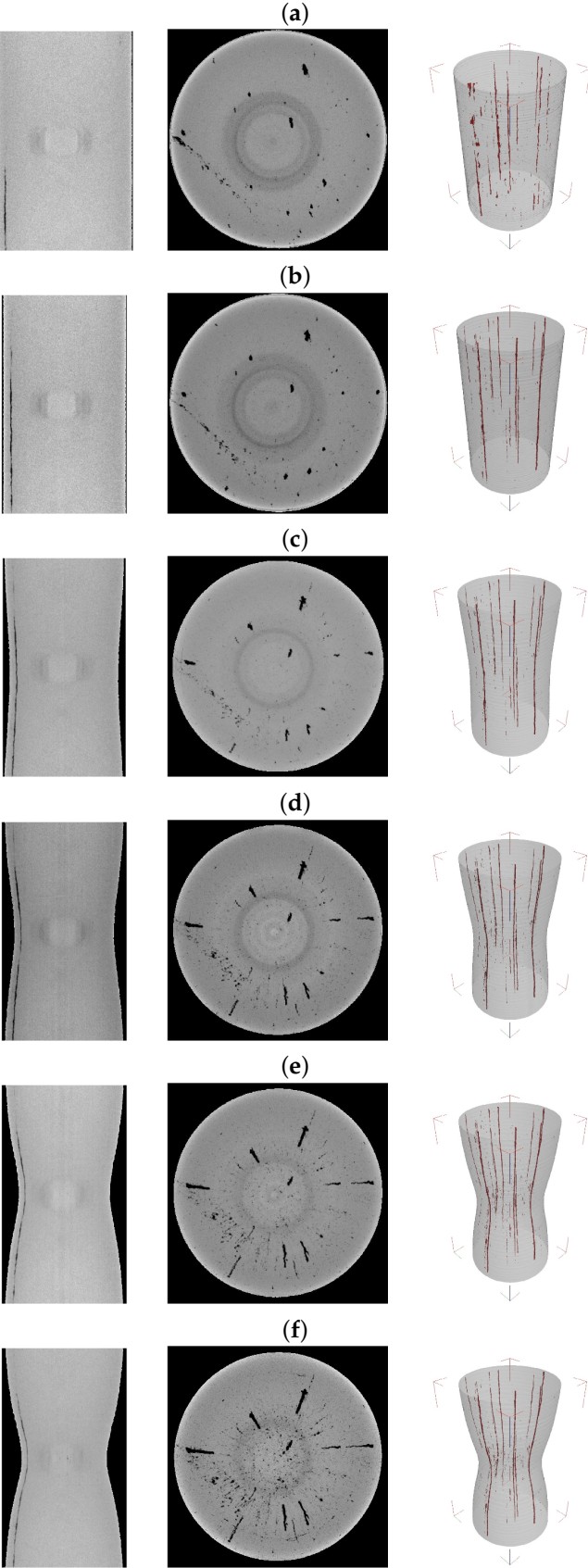

**Figure 10.** Results of the XRCT analysis of the necking zone for Material 2. For each step, three results are given: (I) Longitudinal section, (II) Damage projected on the cross section and (III) Perspective of the internal damage. (**a**) Step 0: Before testing ($\varepsilon_{eng} = 0$). (**b**) Step 1: Maximum load ($\varepsilon_{eng} = 0.193$). (**c**) Step 2 ($\varepsilon_{eng} = 0.267$). (**d**) Step 3 ($\varepsilon_{eng} = 0.288$). (**e**) Step 4 ($\varepsilon_{eng} = 0.306$). (**f**) Step 5 ($\varepsilon_{eng} = 0.319$).

As in the Material 1 specimen, the mechanism of nucleation and growth of voids is clearly noticeable: as the test progresses, evenly spaced voids start to appear and grow.

### 2.4.4. Longitudinal and Radial Distribution of Voids at Each Step

To understand how voids nucleate and grow inside the specimen during the test, their distribution has been obtained in the longitudinal direction and in the radial direction for each step. In the longitudinal direction, the specimen part considered was 3.5 mm long centered at the necking area and the void volume was measured for 0.025 mm-long slices. In the radial direction the volume was measured at seven concentric hollow cylinders (except the smaller one, which was a full cylinder) of the same volume; this measurement was obtained for 349 slices, 0.8725 mm in length. Figure 11 shows schematic diagrams of how these values were obtained; subfigure (a) shows a scheme of one of the hollow cylinders used to compute the void volume fraction (VVF) at a certain radius for the radial distribution and subfigure (b) the scheme of one of the slices used to compute the VVF for a point in the longitudinal distribution. To do this work, the raw data obtained with XRCT was filtered by means of MATLAB® scripts and functions [30].

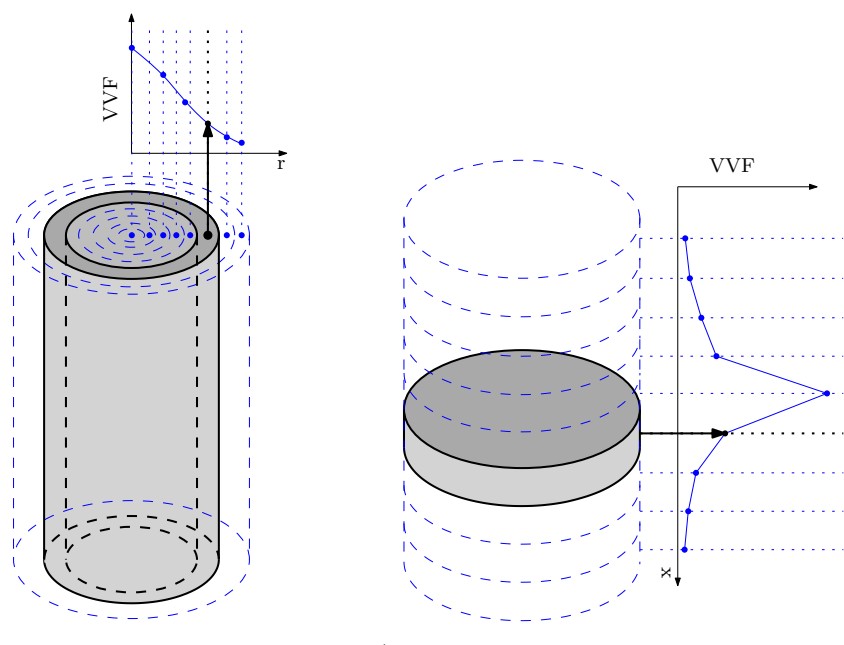

a) Radial distribution of VVF　　　　b) Longitudinal distribution of VVF

**Figure 11.** Schematic representation of the volumes used to obtain a point of the VVF distribution for both directions.

### 2.5. Discussion on the Experimental Data

The fractographs obtained with the 9 mm-diameter specimens (Figures 5 and 6) show two clearly different fracture behaviors between Material 1 and Material 2. While the former presents two clearly defined failure mechanisms, nucleation and growth of microvoids in the internal region and cleavage in the external one, the latter presents a global nucleation, growth, and coalescence mechanism all over the surface and a combination of this mechanism with a slight cleavage only in the very external part of the inclined plane of the cup-cone shape. This last observation agrees with previous research published by Scheider and Brocks [31] or by Suárez et al. [8,9].

Therefore, while the fracture surface of Material 2 and its cup-cone shape agrees with the observations by Bluhm and Morrissey [32] and suggests a clearly ductile behavior along the test, the fracture surface of Material 1 shows a different fracture behavior. It suggests that the internal damage, represented by the internal region, develops progressively and opens an internal crack until

it eventually fails, propagating the internal crack outwards by a cleavage mechanism. Hence, Material 1 failure presents a brittle-ductile transition phenomenon.

As regards the tomographic images, the initial image obtained for both specimens, before testing, highlights the different nature of both materials. While Material 1 presents almost no internal voids, Material 2 shows a high volume of them. These voids are aligned longitudinally, which may be due to the manufacturing process.

The different nature of these materials is also noticed looking at how different their behavior is in terms of strain; Material 1 has a maximum strain of around 0.13 at failure while Material 2 reaches a value of about 0.34. This is also evident given that necking is much more noticeable in Material 2 than in Material 1.

In Material 1, some small voids can be noticed under maximum loading. As strain increases, new voids nucleate, some in a random fashion but others longitudinally aligned. The most interesting remark entails the last step, which is very close to failure. Before this instant, voids appear not depending on each other and not connected to each other, but at step four, internal cracks can be observed, including a large crack in the center of the necking region, which is almost plane. This observation agrees with the hypothesis posed in [8,9,33], where the authors suggested that fracture initiates by an internal crack that, once large enough, acts as an internal notch that triggers a brittle fracture process. This is interesting since it allows the use of certain models from the field of linear elastic fracture mechanics which work reasonably well for a clearly elastic-plastic material, such as those based on the cohesive crack concept [34–36]. For instance, in [9] the cohesive crack was successfully used by means of an interface element to reproduce the tensile test on specimens of Material 1, but other approaches such as the embedded cohesive crack [37–39] or the smeared crack modelling [40–42] could also be proposed.

In the case of Material 2, a high number of voids or inclusions is present before loading. As load increases, the number of voids and their volume also increases. In contrast to the other material, void coalescence cannot be identified. It is unclear if this means that coalescence develops extremely close to the eventual failure or if the test could not be stopped close enough to this instant, when internal cracking could be observed. According to the results reported by Bluhm and Morrisey [32], a fracture plane perpendicular to the loading direction could be expected, which would result in the flat surface of the cup-cone fracture pattern, and finally inclined fracture planes, which would result in the shear planes of the cup-cone shape.

Therefore, the internal damage evolution analysis confirms that in the case of Material 2, failure is due to a generalized weakening process that takes place all over the cross section because of a nucleation, growth, and coalescence of microvoids mechanism. In the case of Material 1, although this mechanism is also observed, the eventual failure of the specimen is provoked by an internal crack opening that takes place in the very late instant of the tensile test. This observation confirms that the failure of Material 1 presents a brittle-ductile transition phenomenon where the specimen shows a ductile behavior until the internal crack is large enough to provoke an eventual brittle failure.

## 3. Numerical Work

This section presents the numerical work carried out in this study. To analyze the numerical behavior of the Gurson model, which is widely used for reproducing the material degradation in metals, the experimental data is compared with numerical simulations using this model. The tensile tests of both materials are reproduced numerically by means of the finite element method, using the implicit version of the commercial software Abaqus® [43].

It must be noted that here the original Gurson model is used, that is to say, the model does not take into account the effect of coalescence, introduced later by means of additional parameters proposed by Tvergaard and Needleman in [4]. The reason of this decision is that calibrating a model including coalescence, therefore using the additional parameters added by Tvergaard and Needleman, results in a higher number of parameters to be defined. Since, as remarked by other researchers (e.g., [44]),

a set of parameters that provide a perfect fit of the macroscopical behavior (i.e., load-strain diagram) does not guarantee that the micromechanical behavior is correctly captured, here, as a first approach to the problem, it is preferred to keep the analysis simpler and limit their number by using the original Gurson model.

### 3.1. Description of the Finite Element Model

#### 3.1.1. Geometry

Because of the axial symmetry of the problem, only 1/24th of each specimen has been considered, as shown in Figure 12. To force necking at the $x = 0$ plane, the specimen is not perfectly cylindrical, but its radius varies from 1.5 mm at $x = 0$ to 1.51 mm at $x = 7.25$ mm, enough to induce stress concentration numerically at $x = 0$. The mesh was defined after a mesh-size convergence study [33] and resulted in a mesh of 141,378 elements shown in Figure 13. The length of the elements in the longitudinal direction in the necking region of the specimen was 0.094 mm, while in the radial and circumferential directions the element sizes were of an average side length of around 0.020 mm. The elements were eight-node brick elements with reduced integration (C3D8R, according to Abaqus® nomenclature).

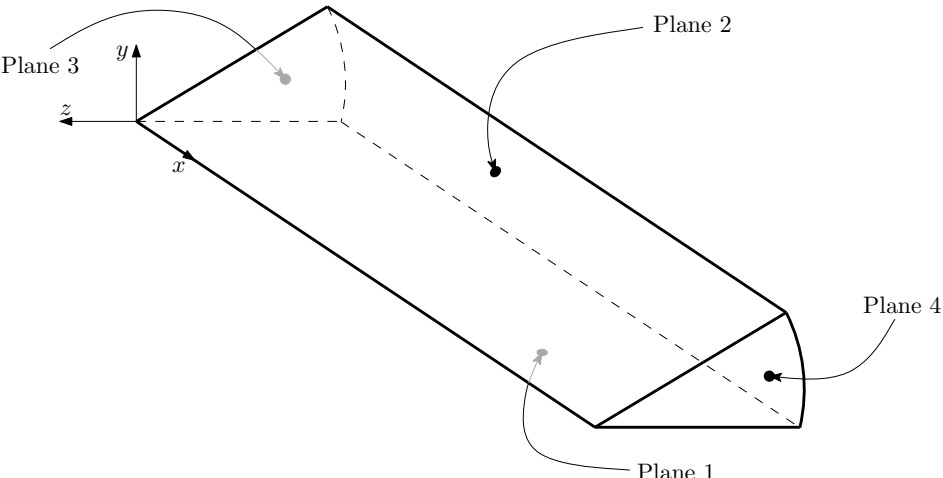

**Figure 12.** Description of the model. Plane 3 represents the eventual plane of failure; displacement is imposed in direction of the $x$ axis on plane 4.

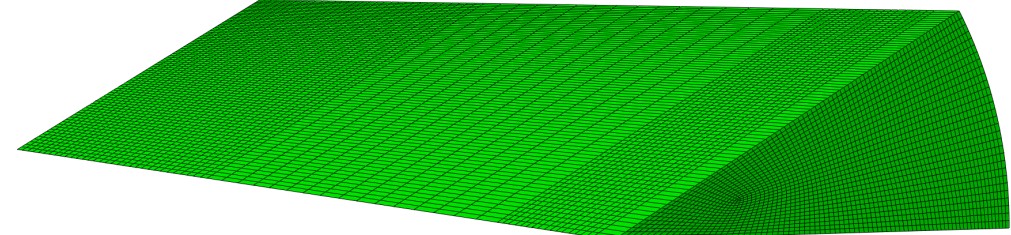

**Figure 13.** Mesh used to reproduce the experimental results.

#### 3.1.2. Boundary Conditions and Load

The load is applied in the X-axis direction and the YZ plane represents the eventual plane of fracture. Regarding the boundary conditions, nodes on the $x = 0$ plane are constrained in the $x$ direction, nodes on the $xz$ plane (plane 1) are constrained only in the $y$ direction and nodes on the inclined plane (plane 2) are constrained only perpendicularly to the plane.

The load is applied by defining an imposed displacement in the $x$ direction to the nodes placed at $x = 7.25$ mm (plane 4).

### 3.1.3. Materials

The specimens are modelled with a porous elastic-plastic material that follows the Gurson's formulation available in Abaqus®.

The elastic-plastic behavior of the latter has been defined by the $\sigma - \varepsilon$ curve obtained experimentally up to the maximum load point. As observed in [8], up to this point the diagram does not depend on the initial length used for measuring strain, therefore, this initial part of the diagram is very well described by the experimental data. The rest of the curve cannot be directly inferred from the experimental data since, in the lab a certain gauge length is used, but at a material level the stress must be related to strain, which is dimensionless. The subsequent hardening slope is in principle unknown and must be estimated. This has been under the assumption of a linear hardening law by using a parameter $r$ which provides the slope after maximum loading as $r = \dfrac{\Delta\sigma}{\Delta\varepsilon}$.

The porous feature is modelled by a Gurson model, where the yield criterion is given by:

$$\Phi = \left(\frac{q}{\sigma_y}\right)^2 + 2f \cosh\left(-\frac{3p}{2\sigma_y}\right) - \left(1 + f^2\right) = 0 \tag{1}$$

where $p$ and $q$ are the hydrostatic pressure and the von Mises equivalent stress, respectively and $f$ is the VVF.

The change in the VVF is computed as the summation of the change due to nucleation ($\dot{f}_{nucl}$) and the change due to the growth of existing microvoids ($\dot{f}_{gr}$) that are obtained by the following expressions:

$$\dot{f}_{gr} = (1 - f)\dot{\varepsilon}^{pl} : \boldsymbol{I}$$

$$\dot{f}_{nucl} = A\dot{\bar{\varepsilon}}_m^{pl}$$

where $\varepsilon^{pl}$ stands for the plastic flow, $\dot{\bar{\varepsilon}}_m^{pl}$ for the equivalent plastic strain in the metal matrix and $A$ is obtained as follows:

$$A = \frac{f_N}{s_N\sqrt{2\pi}} \exp\left[-\frac{1}{2}\left(\frac{\bar{\varepsilon}_m^{pl} - \varepsilon_N}{s_N}\right)^2\right]$$

Therefore, the nucleation strain follows a normal distribution with a mean value $\varepsilon_N$, a standard deviation $s_N$ and $f_N$ stands for the volume fraction of nucleated voids.

### 3.2. Calibrated Models

With the aforementioned description of the model, the following parameters can be identified for calibration:

- Material relative density, $d$. Please note that here we follow the Gurson model parameters used in the implementation of the model available in Abaqus®, therefore a value of $d = 1$ implies a fully dense material with an initial VVF of $f = 0$.
- Hardening slope after the maximum load defined as a stress–strain ratio, $r$.
- Mean equivalent plastic strain for void nucleation, $\varepsilon_N$.
- Standard deviation of the distribution, $s_N$.
- Volumetric fraction of nucleated voids, $f_N$.

To calibrate the model for both materials, the numerical results are compared with the experimental data by means of two criteria. The load-strain curve must be similar enough and the evolution of the necking radius in the center of the neck must follow the same pattern as experimentally observed. Figures 8 and 14 show that both criteria are met for both materials. It is interesting to note that in this calibration parameter $r$ and the specific parameters of the Gurson model ($r$, $\varepsilon_N$, $s_N$ and $f_N$) govern hardening and softening processes, respectively. These processes are overlapped and have an

opposed effect on each other, which makes difficult to distinguish the influence of each on the overall behavior of the element.

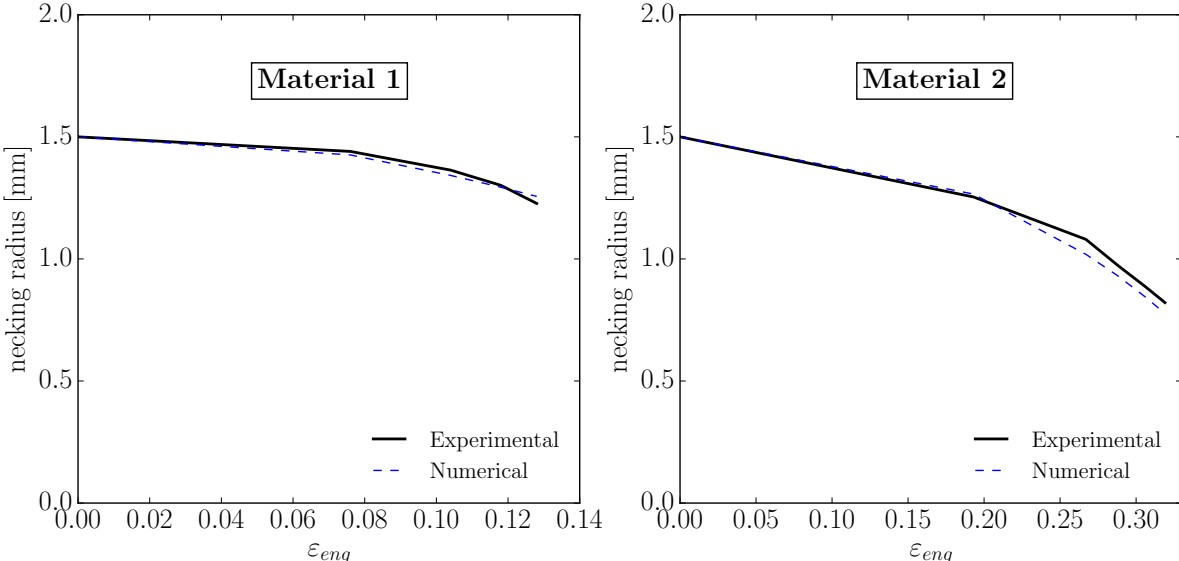

**Figure 14.** Necking radius evolution for Material 1 (left) and Material 2 (right); numerical and experimental results are compared.

The calibration process was carried out by trial and error using around 50 different sets of parameters for each material. As a result, of this process, the sets of parameters shown at Table 3 were obtained. It is interesting to observe that parameter $d$ is much closer to 1 in the case of Material 1 than in the case of Material 2, which agrees well with the experimental observations that prove that the latter has a higher initial porosity compared with the former.

**Table 3.** Initial parameters for FEM models of both materials.

| Material | E [N/mm$^2$] | $\nu$ | $r$ | $d$ | $\varepsilon_N$ | $s_N$ | $f_N$ |
|----------|-------------|------|-----|-------|-------|-------|-------|
| 1 | 160,385 | 0.30 | 782 | 0.999 | 0.4 | 0.1 | 0.02 |
| 2 | 191,536 | 0.30 | 762 | 0.99 | 0.3 | 0.1 | 0.06 |

3.2.1. Comparison with the Experimental Data

The calibration of both models ensures good agreement with the experimental data at the macroscopic level. Nevertheless, the interest in this study is to compare the evolution of porosity inside the material along the tensile test.

The XRCT technique provides information of the interior of the specimen in the form of voxels grouped in slices. That is to say, a slice is the group of voxels placed at the same longitudinal distance from the fracture plane, considered as the origin. To obtain the VVF longitudinal profiles with the experimental data, the porosity fraction is counted for each slice.

Regarding the VVF radial profiles, the void fraction is obtained for concentric cylindrical rings. The inner cylinder is full, and the rest are hollow. Throughout this process, the measurement of internal porosity is carefully obtained, neglecting the internal fracture at step four in Material 1 and the longitudinal porosity chains in Material 2, which appear even in the initial tomography, taken before any load is applied.

The same procedure is followed to obtain the VVF longitudinal and radial profiles with the numerical models; to do this, the results obtained with Abaqus® are filtered by several Python-language scripts using NumPy and SciPy libraries [45–47] and the profiles are extracted for the same strain values considered experimentally (see Table 2). Since the VVF is obtained in Abaqus® at each integration

point, these filters integrate the whole void volume fraction carefully taking into account the tributary volume of each of them and obtain the VVF for any desired volume: hollow cylinders in the case of the radial distribution and cylindrical slices in the case of the longitudinal distribution.

Figures 15 and 16 compare the numerical and experimental profiles. Please note that regarding the experimental results, the diagrams show the measured VVF, which can be slightly lower than the real one, since these results are affected by the equipment precision (2.5 µm). In both materials the longitudinal and radial profiles numerically obtained are somewhat different from the experimental ones.

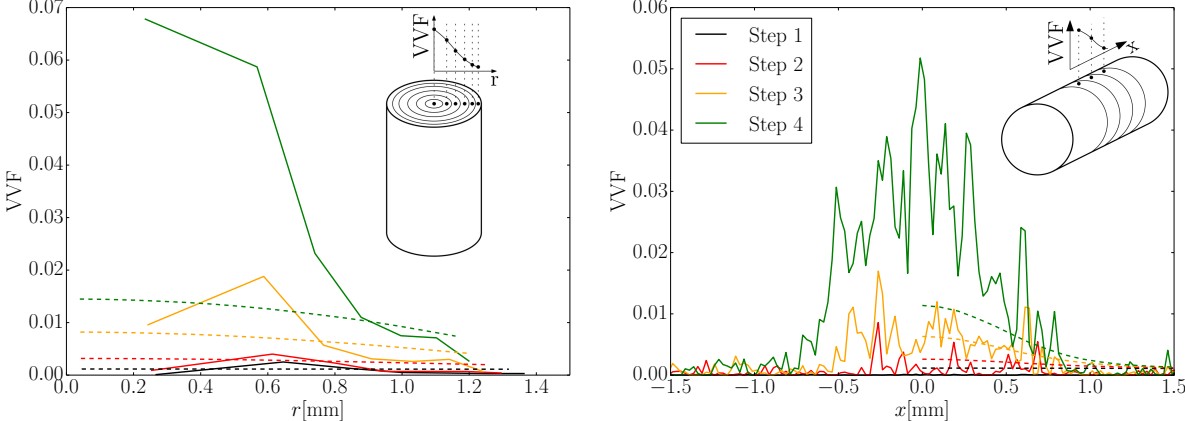

**Figure 15.** Radial (**left**) and longitudinal (**right**) voids distribution experimentally (continuous lines) and numerically (dashed lines) obtained for Material 1.

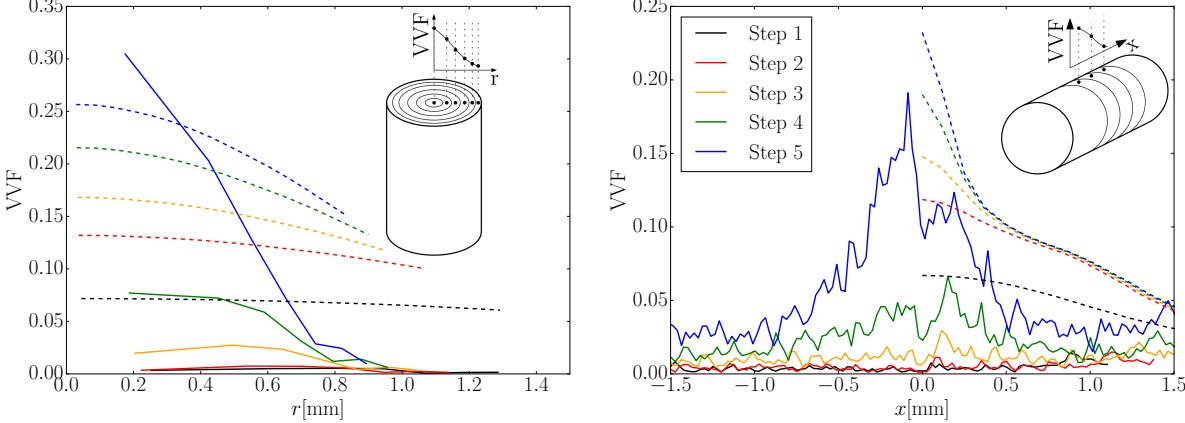

**Figure 16.** Radial (**left**) and longitudinal (**right**) voids distribution experimentally (continuous lines) and numerically (dashed lines) obtained for Material 2.

In the case of Material 1 and regarding the longitudinal profile, these differences might not be so important and, probably, a different adjustment of the model parameters could provide a more similar final profile (green dashed line). Nevertheless, there is an interesting difference, since the experimental data suggest a high porosity development in the last part of the test, between steps 3 and 4; this is not observed in the numerical results. This could be because the model does not reproduce coalescence, but only nucleation and growth of microvoids. Regarding the radial profiles, two important differences can be highlighted: the high porosity increment experimentally observed between steps three and four, which is not observed in the numerical results, and the different shape of the numerical and experimental profiles. The first of these differences can be attributed to the absence of coalescence in the numerical model. Regarding the second difference, the experimental profiles tend to have a steep slope around the middle of the radius, being almost zero at the external part of the specimen, while the numerically obtained profiles tend to be almost flat, even presenting a higher porosity in the external part of the specimen than that measured experimentally.

In the case of Material 2, both radial and longitudinal profiles may seem to be more similar to the experimental ones, since the porosity profiles for the last step are more similar than in the case of Material 1. Nevertheless, by taking a closer look at these profiles the same differences can be pointed out. In the case of the longitudinal profiles, although the last step profile may seem more or less similar to the experimental one, its evolution is different. For example, while the experimental profile does not develop much in the first two steps and only increases slightly in steps three and four, the numerical profile develops much from the very first step. Again, as observed for Material 1, the high porosity increment observed experimentally in the last step is not obtained numerically. In the case of the radial profiles, the same differences observed for Material 1 can be identified, since all the numerical profiles are almost flat, different from the numerical ones, and there is not a high porosity evolution in the last step, as experimentally obtained. Again, as observed previously for the results of Material 1, this last issue may be due to the absence of coalescence in the numerical model.

From these results, it can be concluded that although a set of parameters provides a macroscopically correct response of the specimen, porosity evolution is different not only for Material 1, which could be expected since it is a steel with a little ductile response and a fracture pattern different from the cup-cone shape, but also for Material 2, which in principle is a material with a behavior typically reproduced by Gurson-type models.

Nevertheless, it should be noted that in this work the original Gurson model has been used, thus it only reproduces the effects of nucleation and growth of voids, but not coalescence, which could help finding more similar damage evolution profiles in the last stages of the test.

### 3.2.2. Mesh-Size Effect on the Voids Volume Profiles

Mesh density may have a strong influence on the numerical results that is why, as already mentioned, the model has been calibrated using the load-strain curve and the necking evolution. However, there is still no data about how refining the mesh affects the evolution of volume profiles. To this respect, since the mesh is already fine in the radial and angular directions as can be observed in Figure 13, with sides of around 20 µm in length, only the longitudinal dimension of the elements has been taken into account.

For this study, several meshes have been generated using the same radial and angular discretization and using different element longitudinal lengths in the necking region $l$, ranging from a coarse mesh with $l = 0.75$ mm (Figure 17a) to a fine mesh of $l = 0.075$ mm (Figure 17b). In all cases, the same parameters of the Gurson model have been used for each material, which are those of Table 3.

(**a**)  (**b**)

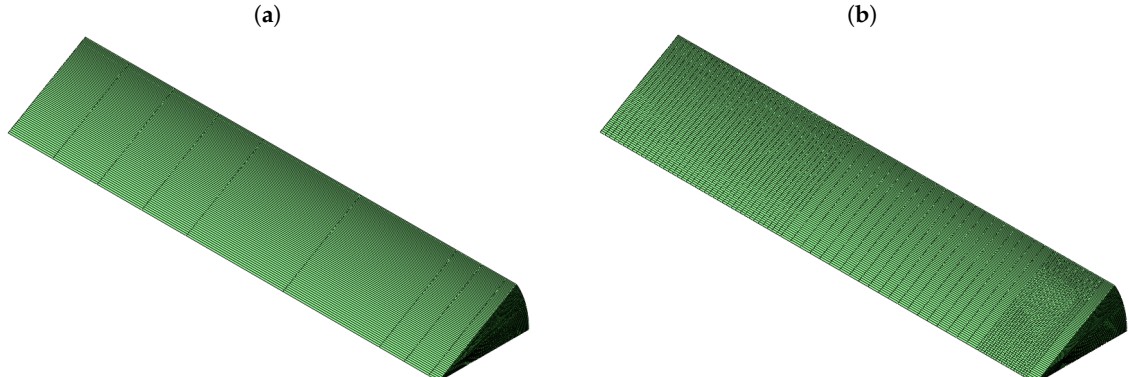

**Figure 17.** (**a**) Coarser mesh with $l = 0.75$ mm and (**b**) Finer mesh with $l = 0.075$ mm.

Figures 18 and 19 show the radial and longitudinal distribution of each mesh at the last considered strain value experimentally measured for each material (step 4 for Material 1 and step 5 for Material 2), which can be considered as the most representative of them all. In the Appendix A at the end of this

article, the reader can find all the voids diagrams for each mesh and for all the strain values considered in this study.

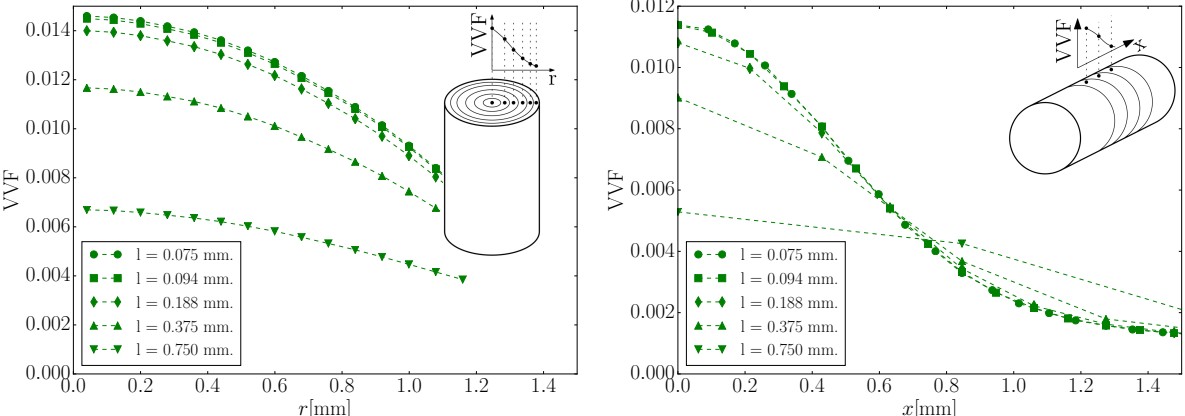

**Figure 18.** Radial (**left**) and longitudinal (**right**) voids distribution of a specimen of Material 1 at step 4 obtained using meshes with different element longitudinal lengths.

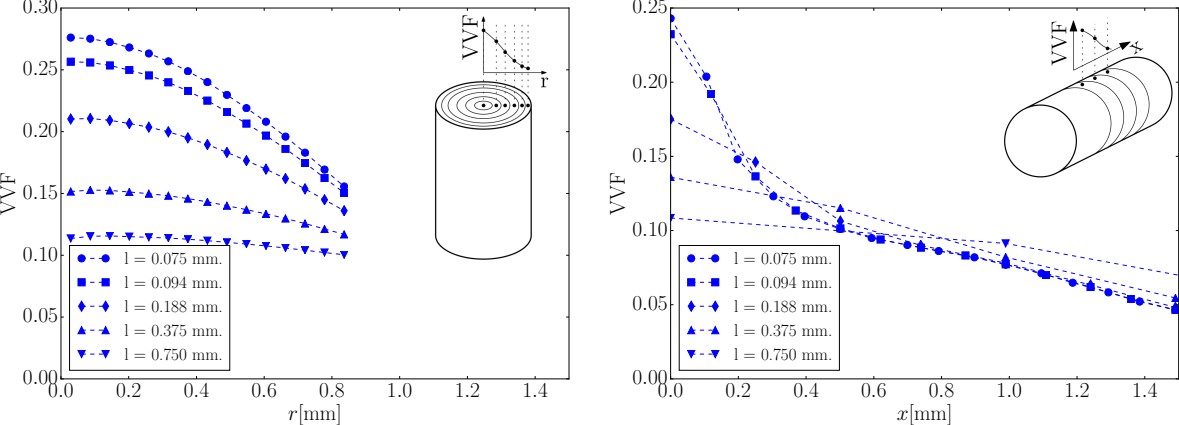

**Figure 19.** Radial (**left**) and longitudinal (**right**) voids distribution of a specimen of Material 2 at step 5 obtained using meshes with different element longitudinal lengths.

In both materials, mesh refinement leads to steeper shapes of both void distributions, radial and longitudinal. Nevertheless, while in the case of Material 1 the second finer mesh ($l = 0.094$ mm.) seems to be fine enough (both diagrams are almost coincident with the finer mesh), in the case of Material 2, it seems that further refinement could lead to slightly steeper shapes of the diagram. This is interesting, since different calibration parameters of the Gurson model seem to require different mesh refinement when the voids distribution shapes are to be analyzed.

In addition to this, the load-strain diagrams are compared in Figure 20, showing that as mesh refines, the load decay at the end of the test is higher, although stabilizes for the finer meshes in both materials. Finally, in Figure 21 the necking evolution is compared showing a similar trend as that observed for the load-strain diagrams, although in this case the deviation for coarser meshes seems to be less important. Since damage, as the experimental observations confirm, is concentrated in a very narrow area of the specimen, coarser meshes cannot correctly reproduce this fact. They extend damage to a wider area, depending on the longitudinal length of the damaged elements and, therefore, for a certain value of $\varepsilon_{eng}$ the load decay in Figure 20 and the necking radius reduction shown in Figure 21 present smaller values, since for a certain strain value a longer part of the specimen is damaged, so damage and necking develop less.

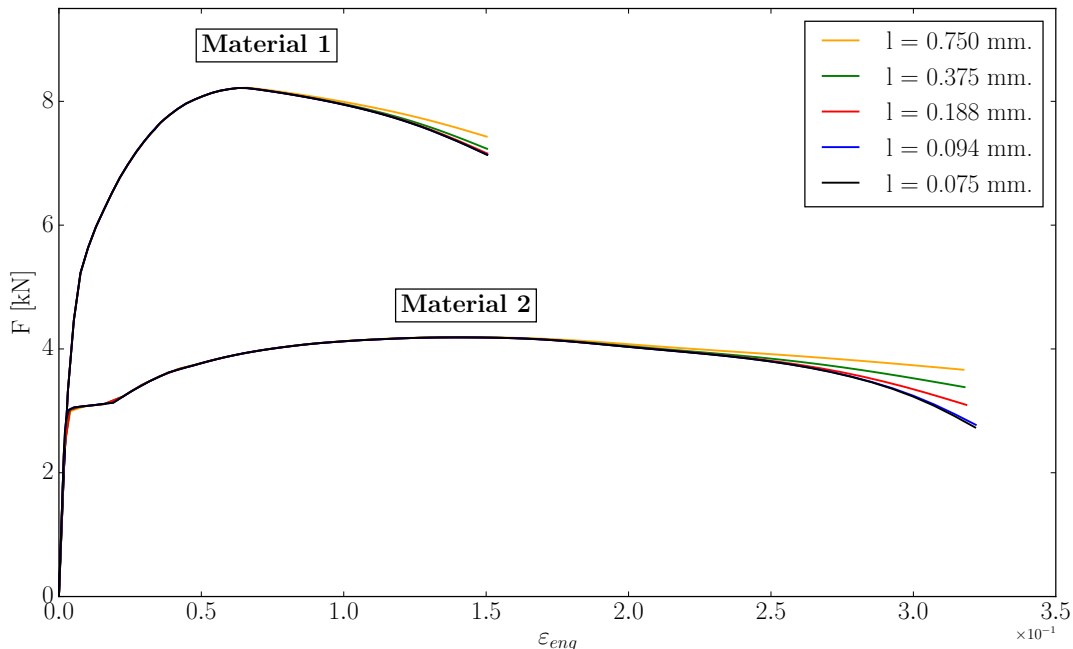

**Figure 20.** Load-strain curves of the 3mm-diameter specimen of both materials; comparison of meshes with different element sizes in the longitudinal direction at the necking region.

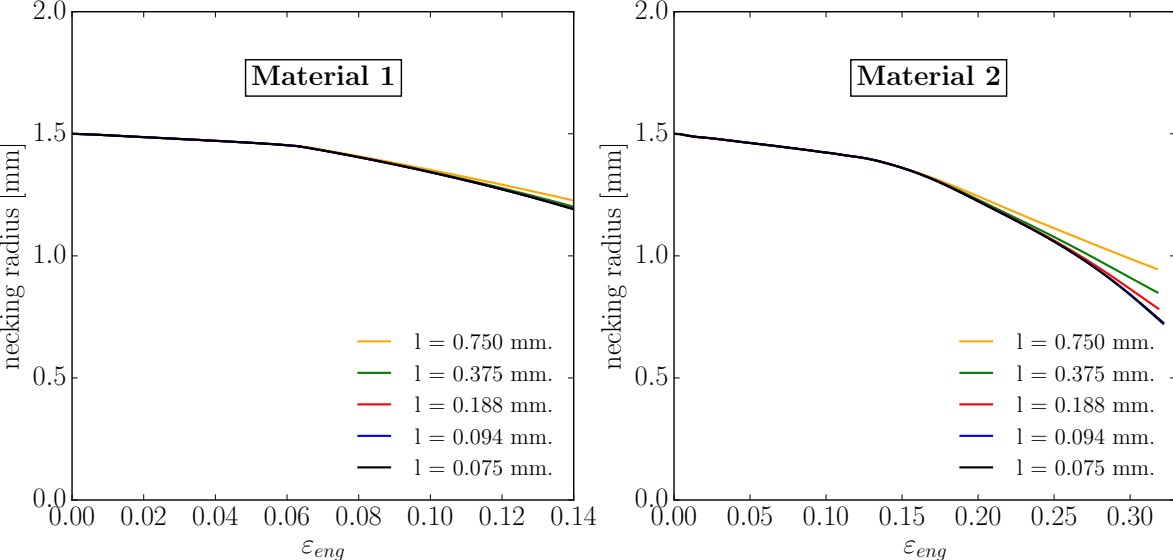

**Figure 21.** Necking radius evolution for Material 1 (**left**) and Material 2 (**right**); comparison of meshes with different element sizes in the longitudinal direction at the necking region.

## 4. Conclusions and Final Remarks

In this paper, two steels with distinct fracture patterns have been analyzed, with Material 1 corresponding to a eutectoid steel used for manufacturing prestressing steel wires and Material 2 being a standard steel used as reinforcement in concrete structures. The study has been carried out on 3 mm-diameter cylindrical specimens by means of tensile test performed in subsequent incremental strain steps up to failure. The internal damage evolution has been identified by means of XRCT.

When a specimen of Material 1 is tested, the fracture surface is plane and perpendicular to the loading direction with two different regions: a central dark region and a brighter surrounding region. In the case of a specimen of Material 2, the fracture pattern corresponds to the cup-cone surface, extensively studied by many researchers.

The fractographic images obtained for both materials allow identification of the mechanism of nucleation, growth, and coalescence of microvoids in each material. Material 1 presents almost no initial voids and a slight internal porosity is developed through the test; the fracture presents a ductile-brittle transition phenomenon. As the test progresses, a ductile behavior is observed where the nucleation and growth of microvoids mechanism is developed and, immediately before failure, a large penny-shaped internal crack, perpendicular to the loading direction, is formed. This leads to an eventual brittle fracture acting as an internal notch, which agrees well with previous works [8,9]. Material 2 has an initial high volume of voids or inclusions, probably due to the manufacturing process, which increases as the tensile test is carried out. In this case, the fracture is evenly developed over the whole cross section: the nucleation and growth of microvoids mechanism takes place evenly throughout the test and causes a progressive weakening of the material that does not provoke a critical internal crack. Although no internal fracture planes have been observed in Material 2, according to Bluhm and Morrisey in [32], the progressive formation of the cup-cone fracture surface should be expected. This could be due to a fast formation of this surface before fracture.

Since the Gurson model allows identification of the development of voids inside the material, the test has been numerically reproduced by means of finite element models by using a Gurson porous material. To calibrate the parameters for each material, two macroscopic values have been considered: the load-strain curve and the evolution of the necking radius. After calibration, internal porosity evolution has been compared with that obtained experimentally for both materials in the longitudinal and radial directions. It is interesting to observe that although the Gurson model can correctly reproduce the two macroscopic criteria used for calibration in both materials, internal porosity profiles differ considerably from the experimental ones. It is also interesting to observe that when the void volume evolution is analyzed, the numerical model calibration may require different mesh refinement for distinct set of calibrated parameters. The use of a complete Gurson-Tvergaard-Needleman model [4], which includes additional material parameters to account for the effects of void interactions and consider the effect of void coalescence, could help to improve the comparison with the experimental results. Nevertheless, it is important to note that the resolution of the XRCT must always be kept in mind, since this experimental technique can underestimate to some extent the material porosity in the material.

As a concluding remark, it is interesting to note that the Gurson model formulation is elegantly based on the microscopic mechanical behavior of spherical voids inside a plastic material, therefore, one would expect to observe a similar void evolution in the experimental results. Nevertheless, the results presented in this work show that although a set of parameters can calibrate the Gurson model and globally reproduce correctly the experimental behavior (load-strain curve and necking evolution) there is clearly need for improved knowledge concerning the calibration of this type of models. In the case that only the macroscopic behavior must be reproduced, there are other possible approaches that can provide good results. For instance, in [9], the authors used the cohesive zone approach with success, which had an advantage respect with the Gurson model: it needs a smaller parameter to calibrate and they can be obtained by standardized tests. Nevertheless, the use of models based on Gurson's formulation, which are based on the micromechanical process of fracture (nucleation and growth of microvoids) is very appealing and interesting. Hopefully, the results presented here may help finding out how to better calibrate these models to match the experimentally observed damage evolution.

**Author Contributions:** J.C.G., J.M.A. and D.A.C. conceived and designed the experimental work and supervised the numerical work; F.S. (Fernando Suárez) carried out the experimental work on the tensile testing; F.S. (Fernando Suárez) carried out the numerical work that deals with the Gurson modelling under the supervision of D.A.C.; F.S. (Federico Sket) and J.M.-A. carried out the XRCT analysis; F.S. (Fernando Suárez) wrote the paper.

**Funding:** This research was funded by the Spanish Ministry of Economy, Industry and Competitiveness by means of the Research Fund Project BIA 2016 78742-C2-2-R.

**Acknowledgments:** The authors want to express their gratitude to Luis del Pozo and Luisa Villares, from Emesa Trefilería, S.A. (Arteixo, La Coruña) for supplying the steel wires, as well as for providing their useful comments.

**Conflicts of Interest:** The authors declare no conflict of interest. The funding sponsors had no role in the design of the study; in the collection, analysis or interpretation of data; in the writing of the manuscript and in the decision to publish the results.

## Appendix A. Voids Volume Evolution Profiles

To give a glimpse on how radial and longitudinal voids evolution profiles are affected by the longitudinal size of the elements in the numerical model, this appendix shows them for each of the strain steps experimentally analyzed using different longitudinal element lengths $l$, ranging from 0.75 mm. to 0.075 mm.

As in previous figures of the article, black lines are used for step 1, red lines for step 2, yellow for step 3, green for step 4 and blue for step 5.

### *Material 1*
(**a**) $l = 0.75$ mm.:

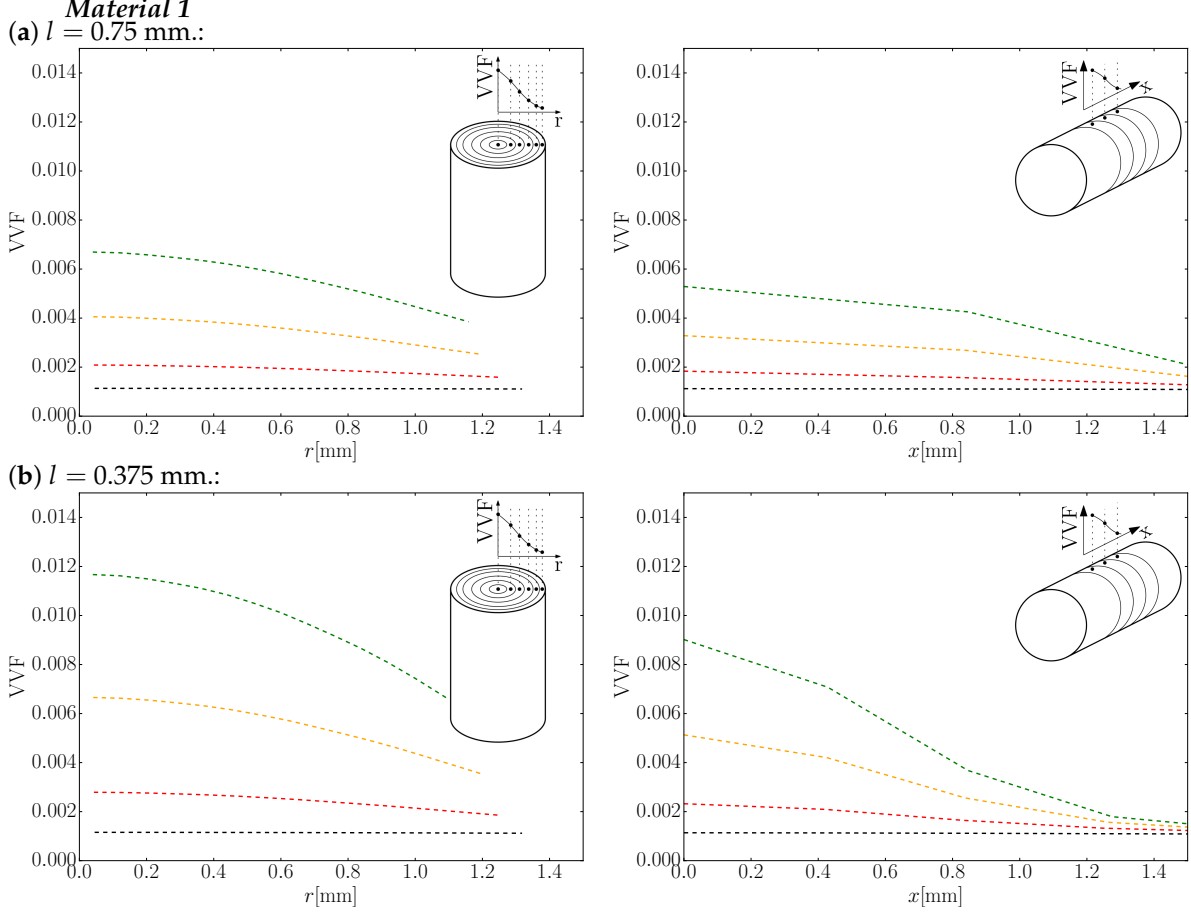

(**b**) $l = 0.375$ mm.:

**(c)** $l = 0.188$ mm.:

**(d)** $l = 0.095$ mm.:

**(e)** $l = 0.075$ mm.:

*Material 2*

**(a)** $l = 0.75$ mm.:

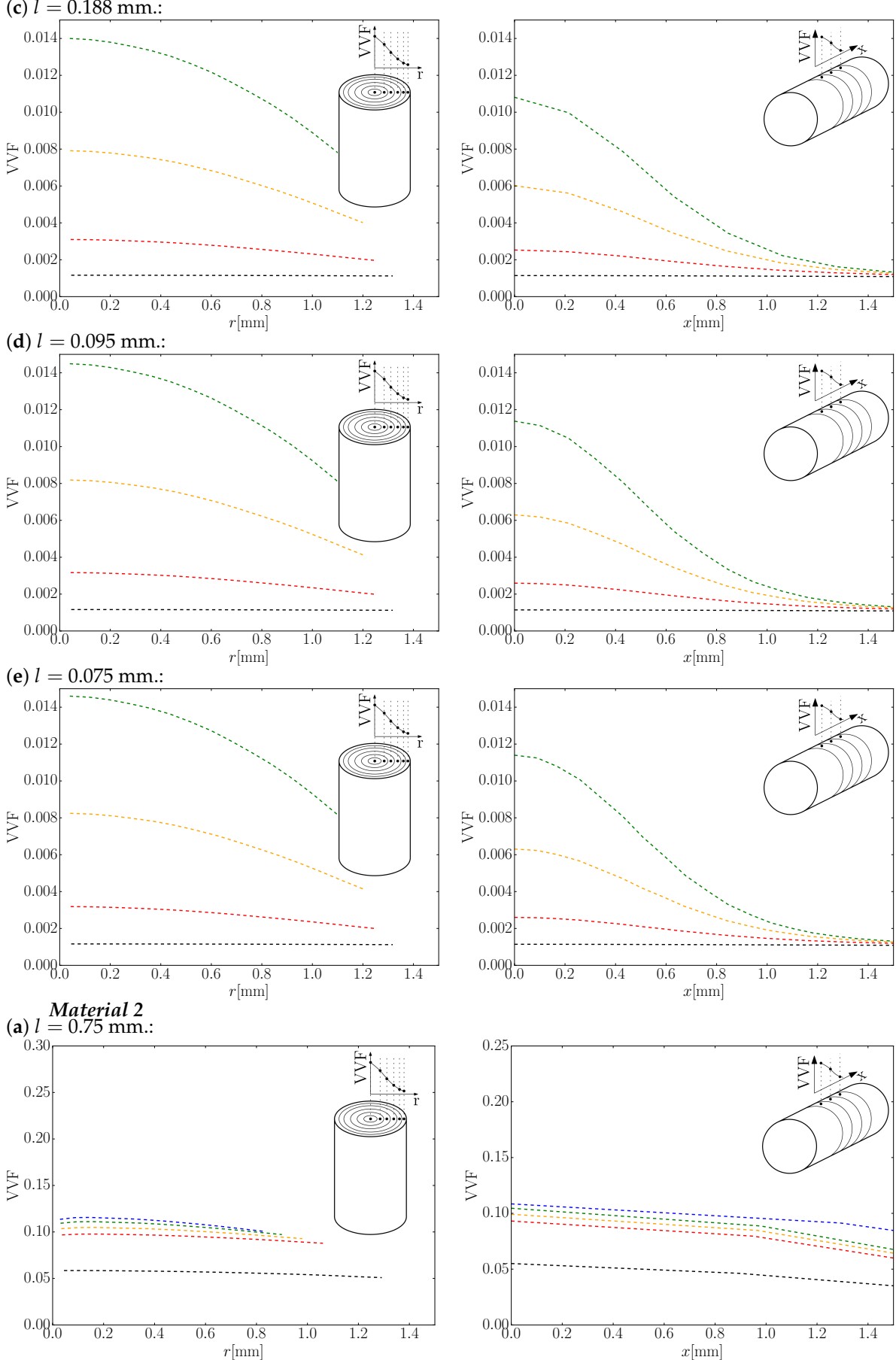

**(b)** $l = 0.375$ mm.:

**(c)** $l = 0.188$ mm.:

**(d)** $l = 0.095$ mm.:

**(e)** $l = 0.075$ mm.:

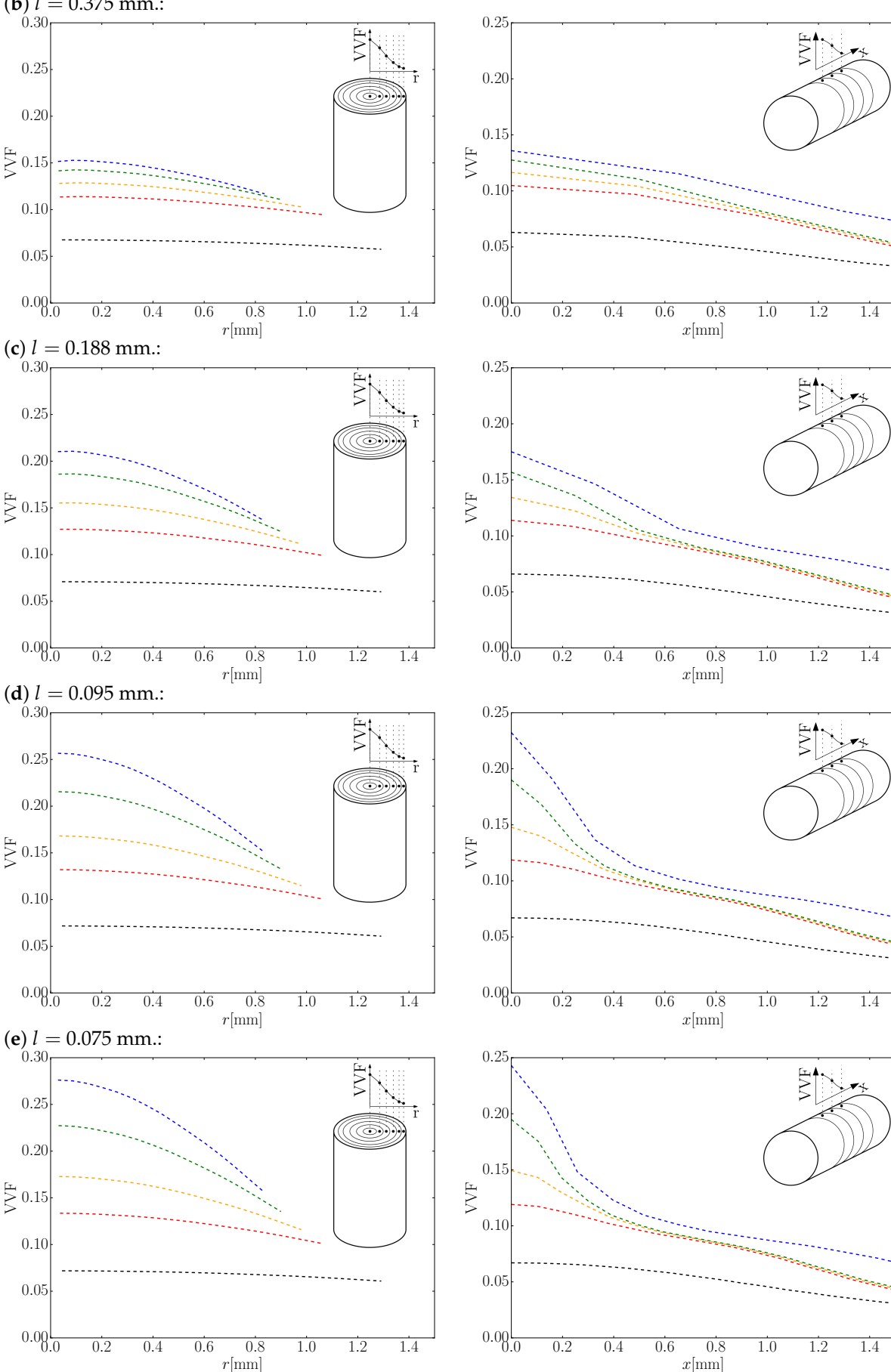

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
