# Peer review of "The Evolution of Internal Damage Identified by Means of X-ray Computed Tomography in Two Steels and the Ensuing Relation with Gurson’s Numerical Modelling"

_metals, doi:10.3390/met9030292_

Reviewer 1 Report

The authors present a study of the evolution of voids in two different steels based on X-ray tomography and other microscopical analyses. In particular, the evolution of voids and fracture surface in uniaxial tension tests are evaluated. The authors calibrate a Gurson model to predict the evolution of voids in the two materials. The study is interesting for readers of Metals and generally well written. However, the reviewer has concerns regarding the methodology and the corresponding conclusions. There is also an issue about the originality of the submitted manuscript. Therefore, a thorough revision of the manuscript in these points is recommended. Please, see the detailed comments.

Author Response

Dear Editor,

Please, find enclosed the updated revised version of the following paper:

Journal:                      Metals

Manuscript ID:           metals-419571

Title:                          The evolution of internal damage identified by means of X-ray computed                                   tomography in two steels and the ensuing relation with Gurson’s numerical                          modelling

Author:                       F. Suárez, F. Sket, J.C. Gálvez, D.A. Cendón, J.M. Atienza, J. Molina-Aldareguia

This revision responds the last reviewer, (Reviewer 1) whose revision was received by us on the 23rdof January, after we sent a former revision responding the other three reviewers. We would like to thank this reviewer for his comments, although we had already addressed several changes thanks to the other reviewers, this has helped us correct some issues and improve the manuscript.

We also respond the second round of Reviewer 2, which has helped us polish the final version, especially with his first comment.

Below we address each of the remarks made by the authors (in blue) to the comments of Reviewer 1 and reference the changes made in the manuscript (highlighted in green in the new version). Note that there are more modifications that correspond to other reviewers’ suggestions, which are highlighted in yellow. Finally, we also include the modifications suggested by Reviewer 2, which are highlighted in blue in the new version.

Yours faithfully,

The authors

Comments to reviewers

Reviewer #1

The authors present a study of the evolution of voids in two different steels based on X-ray tomography and other microscopical analyses. In particular, the evolution of voids and fracture surface in uniaxial tension tests are evaluated. The authors calibrate a Gurson model to predict the evolution of voids in the two materials. The study is interesting for readers of Metals and generally well written. However, the reviewer has concerns regarding the methodology and the corresponding conclusions. Therefore, a revision of the manuscript in these points is recommended.

Background, references

1. p. 17 and eventually other positions: The authors write “ductile-fragile transition”. Probably, “ductile-brittle transition” is meant.

            The reviewer is right, this issue has been corrected in the new version.

2. The authors use the terms “damage”, “voids”, “fracture”, “crack”. Clear definitions should be given. In particular, even though “damage” is part of the title, there ambiguities in the text. The reviewer’s definition of damage would be the decrease of load bearing capacity due to defects (of whatever nature). This implies that damage can only be observed when mechanical test data indicating the changes in mechanical properties is compared with other evidence, e.g. microstructural analysis to detect voids. This should be addressed.

            Around the end of the Introduction, some text has been added to clarify this: 

“It must be noted that in this work the evolution of internal porosity is analysed, but also the appearance of true cracks inside the specimen before the eventual failure. For this reason, the term damage is used as a general concept that refers to voids, which may nucleate and grow, and cracks, which can be the result of coalescence of voids, , both processes weakening the material.”

3. The term “sorbitic” is used to describe appearance of pearlite. The reviewer only knows this in the term of martensite. Short clarification would help.

When martensite is cooled slowly the lamelar structure of perlite (alternating ferrite and cementite) can be clearly observed, but when the cooling process is fast, this structure is blurry and is called sorbitic perlite. See, for example, the definition of ‘Sorbite’ in “A Textbook of Engineering Material and Metallurgy“, by Amandeep Singh Wadhwa and Harvinder Singh Dhaliwal: 

“It is a micro-constituent which consists of ferrite and finely divided cementite and is produced on tempering martensite above approx. 450ºC. The constituent also known as sorbitic pearlite is produced when austenite is cooled at rate slower than that which will yield troostite and faster than that which will yield a pearlitic structure.”

Methods

1. The authors state that the resolution of their X-ray tomography is 2.5 micrometre per voxel. That means that all voids having an extension of less than 2.5 \mu m in one dimension cannot be resolved and are not taking into account into the determination of void volume fraction. The dimensions of voids in steels can be substantially smaller (see e.g. Isik et al. (2016), steel research international 87, DOI: 10.1002/srin.201500483) are in the nanometer range. This depends strongly on the composition of the steels, but in general, the reviewer would consider voids which are detectable with 2.5 micrometre per voxel as already large. The authors mention the drawbacks of using post-mortem analysis with SEM based analysis methods. Some of the drawback can be circumvented by using proper polishing technique. In any case the authors should make a comparison with SEM based techniques on the size of voids. The cited paper by Landron et al. (2013, Acta Materialia) mentions a slightly lower voxel size and some comparisons with alternative observation methods. The pearlitic-ferritic steel investigated by the authors probably has quite small voids in particular in the 1pearlite phase.

            We thank the reviewer for this comment. It is true that the resolution of the XRCT equipment (a voxel size of 2.5 microns in this case) limits the accuracy of the results. In our opinion, the results obtained here are quite reliable for both materials since, although voids smaller than 2.5 microns have not been taken into account, a very high number of them would be necessary to have a strong influence on the total VVF. Nevertheless, this issue was not mentioned properly in the previous version and has now been specifically mentioned in the text. Please, see our comments to remarks 1 and 3 of section Results in this review.

2. ASTM standard for grain size (G=xx) determination should be mentioned.

            The Standard used for this measure (UNE-EN ISO 643:2013) has been referenced

3. The authors use different specimens for X-ray measurement and fractographs. It should be checked whether this changes the fracture behavior in particular when necking starts. When necking starts, the deformation mode will most likely be plane strain tension. However, the point in terms of displacement and time when this starts will to some extent be influenced by specimen geometry. Could be checked with DIC measurement or simulation.

            In the case of Material 1, the specimen diameter had an influence on the aspect of the fracture surface (proportion of the diameter of the dark region respect with the necking diameter), while in the case of Material 2 the fracture surface geometry seemed proportional with the specimen diameter. This was already analysed in [8]:“Study of the last part of the stress-deformation curveof construction steels with distinct fracture patterns” , that is why it is not mentioned here. In any case, this does not affect the study carried out here, since observations on 9mm specimens only were used for better identification of the fracture mechanisms on the fracture surfaces, which are the same in any of the analysed specimens sizes (see reference [8], for example).

4. Some more details of the materials (as concerned microstructure such as phase fraction) could be mentioned in the submitted manuscript. There is additional information in the previous papers by the authors, though.

            Since the materials have been analysed in the past, we think it is better to reference those papers instead of giving the same information again, that is why we include the following sentence in the introduction: 

“These materials have been analysed in the past and their mechanical properties are well known; for further information, the reader is referred to [8] and [9]”

5. What is the accuracy of the load cell’s force measurement? There are deviations between single repetitions of the tensile tests. How exactly were the specimens machined in terms of surface roughness?

The accuracy of the load cell (0.5%) has been included in the text. The surface roughness was not assessed, but was clearly smooth with the naked eye.

6. The constitutive model should be explained in more detail. The Abaqus manual is cited, but the constitutive equations giving meaning to the parameters in Table 3 should be stated explicitly in the text.

            The description of the model has been extended at the end of section 3.1.3 to include the expression of A.

7. The authors calibrated the material parameters by fitting to force displacement curves and the observed necking behavior. For fitting to the force-displacement curves at first a hardening law/relation was assumed and then the Gurson parameters governing softening determined. This way it is difficult to distinguish between softening and hardening. Comment on this is needed.

            We find this remark interesting, since it highlights how difficult it can be to find a set of parameters that correctly reproduce the material degradation. We have included some text in section 3.2 highlighting this.

Results

1. The results of the predicted evolution of void volume fraction with the Gurson model are very interesting. However, one should keep in mind that the determined void volume fraction depends on the used methods as mentioned previously.

            Additional text has been added in section 3.2.1 mentioning this issue.

2. The authors perform a mesh sensitivity study focusing on the predicted numerical results with the Gurson model. The reviewer thinks that this can be shortened and e.g. the review article by Song Cao (International Journal of Material Forming 2017) on damage modelling in cold forming referenced.

            We thank the reviewer for this suggestion, but we think that this section is already quite concise in the body of the manuscript. When we included this section, our aim was to keep it as short as possible in order to not deviate the attention of the reader from the main issues, that is why we extended this part in the Appendix.

3. Fig. 10 shows that there is considerably porosity before performing the tensile test. How accurate is this already existing porosity determined?

            The accuracy is limited by the voxel size of the tomograph, which is 2.5 microns.

4. It is interesting to see that the Gurson model underpredicts radial distribution of void volume fraction, but overpredicts in the longitudinal direction (Figure 15)? This can be attributed to a number of things, the determination of the parameters, the measurement of the void volume fraction. More comments are needed here.

            We had already extended section 3.2.1 attending other reviewers’ suggestions and think that now, with the additional modification included regarding the accuracy of the measurement of VVF, it is improved.

Conclusions

1. The conclusion on the good prediction of the mechanical response with the Gurson model, but difficulties to predict void volume fraction seems most interesting.

            We have extended the conclusions (now named as “Conclusions and final remarks”). We think that now the most interesting remarks are now highlighted, including this one.

2. The conclusion on the consideration of void coalescence seems to be difficult to prove with the presented research. There might be other reasons (mentioned in Results section of review) responsible for this.

            This issue is now mentioned in the conclusions.

Reviewer 2 Report

SUMMARY: The authors present an interesting study concerning the evolution of internal damage in two steels. The study is performed considering experimental and numerical results, resorting to X-ray computed tomography and finite element numerical analysis with the Gurson’s model, respectively. The calibration of the Gurson’s model parameters is performed using an inverse analysis approach, using macroscopic results, i.e. the force-displacement curve and the necking profile. The evolution of the internal damage is compared, highlighting some discrepancies, which can be associated with the fact that the coalescence stage was neglected in the numerical model. Nevertheless, it is interesting to note that this is much more relevant for material 1. The manuscript also emphasizes the fact that the evolution of internal damage is influence by the discretization adopted, which poses difficulties to the use of microscopic results for the calibration of the Gurson’s model parameters.

RECOMMENDATION: Globally, the work is considered original and the results presented are quite interesting. In my opinion, the authors should try to improve the description of the numerical model adopted and the strategies used for the results analysis, in order to give emphasizes to original results. Some comments are presented below to try to contribute to improve the quality of the presentation and discussion.

COMMENT: The authors published some previous works considering the same materials discussed in the manuscript. In reference [8] (Suárez, F.; Gálvez, J.C.; Cendón, D.A.; Atienza, J.M. Study of the last part of the stress-deformation curve of construction steels with distinct fracture patterns. Engineering Fracture Mechanics 2016, 166, 43 – 59.), the authors presented the same results as in Figure 1 and discussed the difficulties inherent to the analysis of the stress-strain curve after the onset of necking. In reference [9] (Suárez, F.; Gálvez, J.C.; Cendón, D.A.; Atienza, J.M. Fracture of eutectoid steel bars under tensile loading: 415 Experimental results and numerical simulation. Engineering Fracture Mechanics 2016, 158, 87 – 105.), the authors presented same results as in Figure 1, but focused the analysis on Material 1, discussing in detail the same results as in Figure 5 and studied the fracture mechanism using the finite element method and the cohesive zone approach. In [9], the authors state that “This approach provides reasonably good agreement with the experimental results. In addition, when compared with other popular models such as the GTN model, it presents certain advantages since it requires a smaller number of parameters to be defined.” Therefore, in my opinion, the authors should give some insight about the pros and cons of each model, i.e. should include some comments about their experience with both models.

Finally, in Suárez, F.; Gálvez, J.C.; Cendón, D.A.; Atienza, J.M. Distinct Fracture Patterns in Construction Steels for Reinforced Concrete under Quasistatic Loading — A Review. Metals 2018, 8, 171., the authors presented the same results as in Figure 1, Figure 5, Figure 6, and part of the results in Figure 8, Figure 9 and Figure 10. I understand that it is important to show same of these results in order to give the proper framework for the following discussion. However, in my opinion, the authors should: (i) simplify the discussion of results previously presented, adding the proper reference; and (ii) include the proper reference for figures that have been previously published.

COMMENT: In page 1, the authors state: “The fracture of ductile materials has usually been explained with the theory of nucleation, growth and coalescence of microvoids [2]. (…) At a second stage, and under higher strain states, these microvoids increase their size (growth) until they become interconnected (coalescence).” In my opinion, the authors should revise the text to clarify that coalescence corresponds to a third stage.

COMMENT: In page 2, the authors state: “This theory has given rise to many mathemathical models, with a special mention to Gurson-type models [3]. Since its appearance, the Gurson model has been applied to reproduce damage evolution in different ductile materials, such as aluminium, copper and steel [4–6].” In my opinion, the authors should revise the text to clarify the difference between Gurson model and Gurson-type models, in order to use them in a coherent manner. Note that references [4-6] correspond to the model usually referred as the GTN. Note that, in my opinion, the use of the Gurson-type model is the main novelty of the manuscript. Therefore, I think that the authors should be particularly careful when presenting the model. In this context, the authors should improve the discussion on the results shown in Figure 1, i.e. please explain why this figure is important, including if it is usual to see any differences in the curves before the onset of necking.

As far as I know, the Gurson model only takes into account the growth and requires the presence of an initial void volume fraction. Thus, one of the first extensions introduced was the void volume fraction rate of change due to nucleation, which can be strain-controlled, as suggested by Gurson and used in the current work. The Gurson-type model implemented in ABAQUS, also considers the void interaction parameters, qi, introduced by Tvergaard, based on finite element unit cell computations. Moreover, in order to model the complete loss of load carrying capacity at a realistic level of the void volume fraction, Tvergaard & Needleman further modified the Gurson's spherical yield criterion to account for the onset of void coalescence leading to final material fracture. Therefore, in my opinion, the authors should carefully revise section “3.1.3. Materials”, in order to properly describe the model considerer and the assumptions made, particularly why was coalescence neglected. Note that for the readers that are more familiar with GTN it is strange that the parameters related with coalescence are not mentioned. In fact, only in page 15, after the results discussion, the authors state that coalescence was neglected.

COMMENT: In page 2, the authors state: “Figure 2 shows the fracture surface of two 9mm-diameter specimens made of different steels. The one on the left shows a typical cup-cone fracture, (…) The one on the right shows a flat fracture surface…” In my opinion, the authors should revise the text because there is some confusion between left and right.

COMMENT: In page 4, the authors state: “In order to follow the damage evolution inside a specimen, it was tested in subsequent load stages, after each of which the specimen was unloaded and its neck analysed with X-ray computed 91 tomography (Nanotom 160NF, Phoenix).” In my opinion, the authors should revise the text to make it clearer. They should also revise the steps given afterwards. In particular, they should explain that the specimen is loaded until the maximum load is reach and only them the test is stopped for the first time. In point 4, the authors state “Specimen is tested until the second stage”. I assume they are referring to the growth stage, but I think it is difficult to know exactly when it starts. Moreover, even using X-ray tomography it is difficult to separate nucleation from growth, and growth from coalescence. Thus, I think it is better to indicate that several loading steps are analysis, after the onset of necking. In this context, in the analysis the authors should consider the use of the term “steps” instead of “stages” to avoid any confusion.

COMMENT: In page 7, the authors state: “In the case of Material 1, raw eutectoid pearlitic steel, no differences can be observed when compared with the microstructure before testing (see Figure 3a). In the case of Material 2, standard steel used as reinforcement in concrete structures, grains are oriented in the longitudinal direction after the test…” In my opinion, the authors should help the reader by indicating the levels of engineering strain that are considered in Figure 7, for each material. Also, in my opinion, a proper comparison of the necking geometry (page 11) should be done for the same level of engineering strain.

COMMENT: In page 8, the authors state: “In the longitudinal direction, the specimen part considered was 3.5mm long centered at the necking area and the void volume was measured for 0.025mm-long slices. In the radial direction the volume was measured at seven concentric hollow cylinders (except the smaller one, which was a full cylinder) of the same volume; this measurement was obtained for 349 slices, 0.8725mm in length. To do this work, the raw data obtained with XRCT was filtered by means of Matlab scripts and functions [19].” In my opinion, the authors should explain these details better. In fact, they should consider adding a scheme to make it easier to understand how the volume of the concentric cylinders was specified.

Also, in page 14, the authors state “The same procedure is followed to obtain the VVF longitudinal and radial profiles with the numerical models; to do this, the results obtained with Abaqus are filtered by several Python-language scripts using NumPy and SciPy libraries [24–26] and the profiles are extracted for the same strain rates considered experimentally (see Table 2).” The void volume fraction is an internal parameter, evaluated for each integration point. Thus, in my opinion, it would be interesting to know more details on how the results are post processed. Moreover, it would be interesting to see the evolution of internal damage predicted in the numerical results, i.e. some figures of the void volume fraction distribution, for instance in the longitudinal section.

COMMENT: In page 11, the authors state: “This is interesting since it allows the use of certain models from the field of linear elastic fracture mechanics which work reasonably well for a clearly elastic-plastic material.” In my opinion, the authors should include proper references for the models.

COMMENT: In page 12, the authors state: “The length of the elements in the longitudinal direction in the necking region of the specimen was 0.094 mm.” In my opinion, the authors should include also the length in the radial and circumferential directions. Moreover, the authors should clearly state the type of element used, in order to clarify the details about the type of integration.

COMMENT: In page 12, the authors state: “The elastic-plastic behaviour of the latter has been defined by the s-e curve obtained experimentally up to the maximum load point,…” In my opinion, the authors could improve the explanation taking into the results discussed in reference [8], where they compare the results obtained with an extensometer with the ones of a DIC system. Also, they should explain if the stress-strain curve was fitted with some hardening law or given as a set of points.

COMMENT: Table 3 presents the Gurson-type model parameters, identified for both materials. In my opinion, the authors should include some comments on the values obtained, since it is clear that d is closer to 1 for material 1, which correlates well with the experimental observations. The standard deviation of the distribution is the same for both materials, but the Volumetric fraction of nucleated voids is lower for material 1, which also seems to be in agreement with the experimental data.

COMMENT: In page 14, the authors state: “Nevertheless, there is an interesting difference, since the experimental data suggest a high porosity development in the last part of the test, between steps 3 and 4; this is not observed in the numerical results.” In my opinion, the authors should revise this statement and others, taking into account the fact that the model neglected the coalescence stage. For material 2, the occurrence of strong plastic deformation in the neck region clearly contributes for the void growth, which seems to be overestimated due to the large initial void volume fraction (see differences between numerical and experimental results for the first steps). In this context, “the material behaviour is globally reproduced by the Gurson model”, but clearly improved knowledge concerning the calibration is required.

COMMENT: In page 15, the authors state: “However, there is still no data about how refining the mesh affects the evolution of volume profiles. To this respect, since the mesh is already pretty fine in the radial and angular directions as can be observed in Figure 12, with sides of around 20 mm in length, only the longitudinal dimension of the elements has been taken into account.” The longitudinal dimension has a strong influence on the necking profile and, consequently, the force-displacement curve. Thus, in my opinion, since the authors are analysing this parameter, the influence on both the necking profile and the force-engineering strain curve should be discussed. Note that, I assume that the mesh size effect is being analysed considering always the same parameters for the Gurson model (Table 3). That is why I think these results are important as well as an improved explanation on how the void volume fraction results were treated. This would also help to understand better the results shown in the Appendix A, particularly for the void volume fraction along the longitudinal section.

SUGESTION: In page 2, replace “mathemathical” by “mathematical”

SUGESTION: In page 3, replace “damage evolution, both tests are numerically reproduced.” by “damage evolution, the tensile test for both materials is numerically reproduced.”

SUGESTION: In page 3, replace “elastic limit ft of” by “elastic limit of”

SUGESTION: In page 13, replace “However, the porous feature is modelled by a Gurson model, where the yield criterion is given by the expression (1).” by “The porous feature is modelled by a Gurson model, where the yield criterion is given by:”

SUGESTION: In page 13, replace “In order to calibrate both models,” by “In order to calibrate the model for both materials,”

SUGESTION: In page 13, replace “met for both models.” by “met for both materials.”

SUGESTION: In page 14, replace “the same strain rates considered” by “the same strain values considered”

SUGESTION: In page 15, replace “last considered strain rate experimentally” by “last considered strain value experimentally”

SUGESTION: In page 16, replace “0.075mm..” by “0.075mm.”

SUGESTION: In page 16, replace “all the strain rates considered” by “all the strain values considered”

Author Response

SUMMARY: The authors present an interesting study concerning the evolution of internal damage in two steels. The study is performed considering experimental and numerical results, resorting to X-ray computed tomography and finite element numerical analysis with the Gurson’s model, respectively. The calibration of the Gurson’s model parameters is performed using an inverse analysis approach, using macroscopic results, i.e. the force-displacement curve and the necking profile. The evolution of the internal damage is compared, highlighting some discrepancies, which can be associated with the fact that the coalescence stage was neglected in the numerical model. Nevertheless, it is interesting to note that this is much more relevant for material 1. The manuscript also emphasizes the fact that the evolution of internal damage is influence by the discretization adopted, which poses difficulties to the use of microscopic results for the calibration of the Gurson’s model parameters.

RECOMMENDATION: Globally, the work is considered original and the results presented are quite interesting. In my opinion, the authors should try to improve the description of the numerical model adopted and the strategies used for the results analysis, in order to give emphasizes to original results. Some comments are presented below to try to contribute to improve the quality of the presentation and discussion.

COMMENT: The authors published some previous works considering the same materials discussed in the manuscript. In reference [8] (Suárez, F.; Gálvez, J.C.; Cendón, D.A.; Atienza, J.M. Study of the last part of the stress-deformation curve of construction steels with distinct fracture patterns. Engineering Fracture Mechanics 2016, 166, 43 – 59.), the authors presented the same results as in Figure 1 and discussed the difficulties inherent to the analysis of the stress-strain curve after the onset of necking. In reference [9] (Suárez, F.; Gálvez, J.C.; Cendón, D.A.; Atienza, J.M. Fracture of eutectoid steel bars under tensile loading: 415 Experimental results and numerical simulation. Engineering Fracture Mechanics 2016, 158, 87 – 105.), the authors presented same results as in Figure 1, but focused the analysis on Material 1, discussing in detail the same results as in Figure 5 and studied the fracture mechanism using the finite element method and the cohesive zone approach. In [9], the authors state that “This approach provides reasonably good agreement with the experimental results. In addition, when compared with other popular models such as the GTN model, it presents certain advantages since it requires a smaller number of parameters to be defined.” Therefore, in my opinion, the authors should give some insight about the pros and cons of each model, i.e. should include some comments about their experience with both models.

            A new paragraph has been included at the end of the last section (now called “Conclusions and final remarks”), in this paragraph, apart from addressing another suggestion that will be commented later, this issue is commented and the corresponding paper referenced.

Finally, in Suárez, F.; Gálvez, J.C.; Cendón, D.A.; Atienza, J.M. Distinct Fracture Patterns in Construction Steels for Reinforced Concrete under Quasistatic Loading — A Review. Metals 2018, 8, 171., the authors presented the same results as in Figure 1, Figure 5, Figure 6, and part of the results in Figure 8, Figure 9 and Figure 10. I understand that it is important to show same of these results in order to give the proper framework for the following discussion. However, in my opinion, the authors should: (i) simplify the discussion of results previously presented, adding the proper reference; and (ii) include the proper reference for figures that have been previously published.

            Following the reviewer’s suggestion, we have cited all three previous papers of this research topic and:

       Regarding Figure 1: we are not sure about this reviewers’s comment. We had not published this figure before. In the reference “Study of the last part ...” there is a Figure that may look similar (Fig. 4 in that paper), but in that case the comparison was between two experimental devices, DIC and a traditional extensometer, and here Figure 1 shows that the effect of Gurson’s formulation affects the load-strain curve from very early.

       Regarding Figures 5 and 6: we have removed the general views of the SEM analysis of both materials, leaving only the fractographs that allow identifying distinct fracture mechanisms, which are commented later in the paper.

       Regarding Figures 8, 9 and 10: we have included a sentence at the beginning of section 2.4.3, where the reader is warned that some of these results in these figures were already included in one of the references, but are now completed with more images to better understand the damage process in each material.

COMMENT: In page 1, the authors state: “The fracture of ductile materials has usually been explained with the theory of nucleation, growth and coalescence of microvoids [2]. (…) At a second stage, and under higher strain states, these microvoids increase their size (growth) until they become interconnected (coalescence).” In my opinion, the authors should revise the text to clarify that coalescence corresponds to a third stage.

            This sentence has been modified to clearly describe coalescence as the third stage of the process.

COMMENT: In page 2, the authors state: “This theory has given rise to many mathemathical models, with a special mention to Gurson-type models [3]. Since its appearance, the Gurson model has been applied to reproduce damage evolution in different ductile materials, such as aluminium, copper and steel [4–6].” In my opinion, the authors should revise the text to clarify the difference between Gurson model and Gurson-type models, in order to use them in a coherent manner. Note that references [4-6] correspond to the model usually referred as the GTN. Note that, in my opinion, the use of the Gurson-type model is the main novelty of the manuscript. Therefore, I think that the authors should be particularly careful when presenting the model. In this context, the authors should improve the discussion on the results shown in Figure 1, i.e. please explain why this figure is important, including if it is usual to see any differences in the curves before the onset of necking.

            The paragraph has been rewritten in order to make clearer what we call Gurson-type models.

As far as I know, the Gurson model only takes into account the growth and requires the presence of an initial void volume fraction. Thus, one of the first extensions introduced was the void volume fraction rate of change due to nucleation, which can be strain-controlled, as suggested by Gurson and used in the current work. The Gurson-type model implemented in ABAQUS, also considers the void interaction parameters, qi, introduced by Tvergaard, based on finite element unit cell computations. Moreover, in order to model the complete loss of load carrying capacity at a realistic level of the void volume fraction, Tvergaard & Needleman further modified the Gurson's spherical yield criterion to account for the onset of void coalescence leading to final material fracture. Therefore, in my opinion, the authors should carefully revise section “3.1.3. Materials”, in order to properly describe the model considerer and the assumptions made, particularly why was coalescence neglected. Note that for the readers that are more familiar with GTN it is strange that the parameters related with coalescence are not mentioned. In fact, only in page 15, after the results discussion, the authors state that coalescence was neglected.

            At the beginning of section 3, a new paragraph has been included clarifying this issue:

“It must be noted that here the original Gurson model is used, that is to say, the model does not take into account the effect of coalescence, introduced later by means of additional parameters proposed by Tvergaard and Needleman in []. The reason of this decision is that calibrating a model including coalescence, therefore using the additional parameters added by Tvergaard and Needleman, results in a higher number of parameters to be defined. Since, as remarked by other researchers (e.g. \cite{Steglich1999404}), a set of parameters that provide a perfect fit of the macroscopical behaviour (i.e. load-strain diagram) does not guarantee that the micromechanical behaviour is correctly captured, here, as a first approach to the problem, it is preferred to keep the analysis simpler and limit their number by using the original Gurson model. “

COMMENT: In page 2, the authors state: “Figure 2 shows the fracture surface of two 9mm-diameter specimens made of different steels. The one on the left shows a typical cup-cone fracture, (…) The one on the right shows a flat fracture surface…” In my opinion, the authors should revise the text because there is some confusion between left and right.

The reviewer is absolutely right, this confusion has been amended in the text.

COMMENT: In page 4, the authors state: “In order to follow the damage evolution inside a specimen, it was tested in subsequent load stages, after each of which the specimen was unloaded and its neck analysed with X-ray computed 91 tomography (Nanotom 160NF, Phoenix).” In my opinion, the authors should revise the text to make it clearer. They should also revise the steps given afterwards. In particular, they should explain that the specimen is loaded until the maximum load is reach and only them the test is stopped for the first time. In point 4, the authors state “Specimen is tested until the second stage”. I assume they are referring to the growth stage, but I think it is difficult to know exactly when it starts. Moreover, even using X-ray tomography it is difficult to separate nucleation from growth, and growth from coalescence. Thus, I think it is better to indicate that several loading steps are analysis, after the onset of necking. In this context, in the analysis the authors should consider the use of the term “steps” instead of “stages” to avoid any confusion.

            The description of the testing procedure in section 2.3 has been slightly modified to clarify these issues.

            In regards to the term “stage”, it has been substituted by “step” in order to make things clearer.

COMMENT: In page 7, the authors state: “In the case of Material 1, raw eutectoid pearlitic steel, no differences can be observed when compared with the microstructure before testing (see Figure 3a). In the case of Material 2, standard steel used as reinforcement in concrete structures, grains are oriented in the longitudinal direction after the test…” In my opinion, the authors should help the reader by indicating the levels of engineering strain that are considered in Figure 7, for each material. Also, in my opinion, a proper comparison of the necking geometry (page 11) should be done for the same level of engineering strain.

            The engineering strains that correspond to the microstructures shown in Figure 7 are approximately the strains at the last instant of each test (if we do not count the strain due to elastic deformation, which can be considered as negligible in these materials). This information has been included in Figure 7.

COMMENT: In page 8, the authors state: “In the longitudinal direction, the specimen part considered was 3.5mm long centered at the necking area and the void volume was measured for 0.025mm-long slices. In the radial direction the volume was measured at seven concentric hollow cylinders (except the smaller one, which was a full cylinder) of the same volume; this measurement was obtained for 349 slices, 0.8725mm in length. To do this work, the raw data obtained with XRCT was filtered by means of Matlab scripts and functions [19].” In my opinion, the authors should explain these details better. In fact, they should consider adding a scheme to make it easier to understand how the volume of the concentric cylinders was specified.

            Some text and a new Figure have been added in section 2.4.4 to make this issue clearer.

Also, in page 14, the authors state “The same procedure is followed to obtain the VVF longitudinal and radial profiles with the numerical models; to do this, the results obtained with Abaqus are filtered by several Python-language scripts using NumPy and SciPy libraries [24–26] and the profiles are extracted for the same strain rates considered experimentally (see Table 2).” The void volume fraction is an internal parameter, evaluated for each integration point. Thus, in my opinion, it would be interesting to know more details on how the results are post processed. Moreover, it would be interesting to see the evolution of internal damage predicted in the numerical results, i.e. some figures of the void volume fraction distribution, for instance in the longitudinal section.

            As the reviewer says, the VVF is an internal parameter provided by Abaqus at an integration point level, thus, to obtain the VVF profiles shown in the manuscript, the tributary volume of each integration point must be carefully taken into account. Once this information is known, it is pretty straightforward to obtain the VVF for any volume needed (hollow cylinders in the case of the radial distribution and cylindrical slices in the case of the longitudinal distribution), a simple average must be obtained:

i(VVFi·Volumei)/∑iVolumei

Some text has been included in the mentioned paragraph explaining this.

COMMENT: In page 11, the authors state: “This is interesting since it allows the use of certain models from the field of linear elastic fracture mechanics which work reasonably well for a clearly elastic-plastic material.” In my opinion, the authors should include proper references for the models.

            Some text has been added in this paragraph to mention specific models and the corresponding references have been included.

COMMENT: In page 12, the authors state: “The length of the elements in the longitudinal direction in the necking region of the specimen was 0.094 mm.” In my opinion, the authors should include also the length in the radial and circumferential directions. Moreover, the authors should clearly state the type of element used, in order to clarify the details about the type of integration.

            These issues have been clarified in section 3.1.1.

COMMENT: In page 12, the authors state: “The elastic-plastic behaviour of the latter has been defined by the s-e curve obtained experimentally up to the maximum load point,…” In my opinion, the authors could improve the explanation taking into the results discussed in reference [8], where they compare the results obtained with an extensometer with the ones of a DIC system. Also, they should explain if the stress-strain curve was fitted with some hardening law or given as a set of points.

            Additional text has been added in section 3.1.3, including a citation to the mentioned reference and clarifying the issue about the hardening law, which has been considered as linear; the text now mentions this explicitly for clarification.

COMMENT: Table 3 presents the Gurson-type model parameters, identified for both materials. In my opinion, the authors should include some comments on the values obtained, since it is clear that d is closer to 1 for material 1, which correlates well with the experimental observations. The standard deviation of the distribution is the same for both materials, but the Volumetric fraction of nucleated voids is lower for material 1, which also seems to be in agreement with the experimental data.

            Additional text has been included just before section 3.2.1, where Table 3 is mentioned.

COMMENT: In page 14, the authors state: “Nevertheless, there is an interesting difference, since the experimental data suggest a high porosity development in the last part of the test, between steps 3 and 4; this is not observed in the numerical results.” In my opinion, the authors should revise this statement and others, taking into account the fact that the model neglected the coalescence stage. For material 2, the occurrence of strong plastic deformation in the neck region clearly contributes for the void growth, which seems to be overestimated due to the large initial void volume fraction (see differences between numerical and experimental results for the first steps). In this context, “the material behaviour is globally reproduced by the Gurson model”, but clearly improved knowledge concerning the calibration is required.

            Thank you for this comment. We also think that explaining these differences with the absence of coalescence in the model is important. The text in two paragraphs of the section 3.2.1 has been modified in this sense.

            Finally, a final remark has been included in the Conclusions to highlight this idea, which is in fact one of the key points of the manuscript.

COMMENT: In page 15, the authors state: “However, there is still no data about how refining the mesh affects the evolution of volume profiles. To this respect, since the mesh is already pretty fine in the radial and angular directions as can be observed in Figure 12, with sides of around 20 mm in length, only the longitudinal dimension of the elements has been taken into account.” The longitudinal dimension has a strong influence on the necking profile and, consequently, the force-displacement curve. Thus, in my opinion, since the authors are analysing this parameter, the influence on both the necking profile and the force-engineering strain curve should be discussed. Note that, I assume that the mesh size effect is being analysed considering always the same parameters for the Gurson model (Table 3). That is why I think these results are important as well as an improved explanation on how the void volume fraction results were treated. This would also help to understand better the results shown in the Appendix A, particularly for the void volume fraction along the longitudinal section.

            Two new figures have been included to show how longitudinally refining the elements size affects the load-strain diagrams and the necking radius evolution. Some text have been included before conclusions regarding this new figures and a new paragraph commenting on how they relate each other and the longitudinal and transverse damage profiles has been included.

SUGESTION: In page 2, replace “mathemathical” by “mathematical”

            This is corrected in the new version.

SUGESTION: In page 3, replace “damage evolution, both tests are numerically reproduced.” by “damage evolution, the tensile test for both materials is numerically reproduced.”

            The text has been modified attending this suggestion, it is more accurate now.

SUGESTION: In page 3, replace “elastic limit ft of” by “elastic limit of”

            This is corrected in the new version.

SUGESTION: In page 13, replace “However, the porous feature is modelled by a Gurson model, where the yield criterion is given by the expression (1).” by “The porous feature is modelled by a Gurson model, where the yield criterion is given by:”

            The text has been modified as suggested.

SUGESTION: In page 13, replace “In order to calibrate both models,” by “In order to calibrate the model for both materials,”

            The text has been modified as suggested, it is more precise to talk about one model for two materials, rather than two models.

SUGESTION: In page 13, replace “met for both models.” by “met for both materials.”

            Modified as suggested.

SUGESTION: In page 14, replace “the same strain rates considered” by “the same strain values considered”

            Modified as suggested. The term “rate” was not correctly employed in the first version.

SUGESTION: In page 15, replace “last considered strain rate experimentally” by “last considered strain value experimentally”

            Modified as suggested.

SUGESTION: In page 16, replace “0.075mm..” by “0.075mm.”

            Modified as suggested.

SUGESTION: In page 16, replace “all the strain rates considered” by “all the strain values considered”

            Modified as suggested.

Reviewer 3 Report

A good and interesting article. Suitable research methods. Interesting results, important for designers.

Author Response

The authors thank the kind comments received from the reviewer.

Reviewer 4 Report

This paper entitled "The evolution of internal damage identified by means of X-ray computed tomography in two steels and the ensuing relation with Gurson’s numerical modelling", the authors investigate the internal damage in two types of carbon steel that show different fracture behaviours at the same time that numerical simulations are performed to describe the material’s behaviour.  The topic could be of interest to the Journal “Metals” readers to understand the load-strain curve and the necking radius evolution in both configurations. However, the present manuscript cannot be accepted in its present from due to the following reasons: 1-Frist of all, the manuscript requires English modifications and rewriting large chunks of sentences to shorten them and make them easier to ready by general and scientific viewers. An example of this comment: “This paper analyses the evolution of the internal damage in two types of steel that show different fracture behaviours, with one of them being the initial material used for manufacturing prestressing steel wires, which shows a flat fracture surface perpendicular to the loading direction, and the other one being a standard steel used in reinforced concrete structures, which shows the typical cup-cone surface.” 2-As the topic is not very original, it is necessary to a really updated introduction section with novel publications to describe the state of art of this topic. Additionally, the most recent articles are published by the same authors which present similar figures and results that the one presented in this present work: - Suárez, F.; Gálvez, J.C.; Cendón, D.A.; Atienza, J.M. Study of the last part of the stress-deformation curve of construction steels with distinct fracture patterns. Engineering Fracture Mechanics 2016, 166, 43 – 59. doi: http://dx.doi.org/10.1016/j.engfracmech.2016.08.022. - Suárez, F.; Gálvez, J.C.; Cendón, D.A.; Atienza, J.M. Fracture of eutectoid steel bars under tensile loading: Experimental results and numerical simulation. Engineering Fracture Mechanics 2016, 158, 87 – 105. doi: http://dx.doi.org/10.1016/j.engfracmech.2016.02.044. 3- For the numerical model is necessary to describe how it is considered the porosity in the material characterization. How could affect the porosity in the mechanical properties?. 4- There is a lack of discussion of the presented results with other research papers to compared the results and state what novelty is bringing this research work and how can be used in future for other researcher. 5- The conclusion section is coarse and some paragraphs, like the last three, are discussion instead of conclusions bullets. It is necessary to explicitly described the originality of this work with respect to other research works presented in the same research topic and how can it help this research to the industry.

Author Response

This paper entitled "The evolution of internal damage identified by means of X-ray computed tomography in two steels and the ensuing relation with Gurson’s numerical modelling", the authors investigate the internal damage in two types of carbon steel that show different fracture behaviours at the same time that numerical simulations are performed to describe the material’s behaviour. The topic could be of interest to the Journal “Metals” readers to understand the load-strain curve and the necking radius evolution in both configurations. However, the present manuscript cannot be accepted in its present from due to the following reasons:

1-Frist of all, the manuscript requires English modifications and rewriting large chunks of sentences to shorten them and make them easier to ready by general and scientific viewers. An example of this comment: “This paper analyses the evolution of the internal damage in two types of steel that show different fracture behaviours, with one of them being the initial material used for manufacturing prestressing steel wires, which shows a flat fracture surface perpendicular to the loading direction, and the other one being a standard steel used in reinforced concrete structures, which shows the typical cup-cone surface.”

            Thank you for this recommendation. This sentence has been split into two to make it more readable. Also, the paper has been reviewed in order to modify other long sentences that have been rewritten for readability.

2-As the topic is not very original, it is necessary to a really updated introduction section with novel publications to describe the state of art of this topic. Additionally, the most recent articles are published by the same authors which present similar figures and results that the one presented in this present work: - Suárez, F.; Gálvez, J.C.; Cendón, D.A.; Atienza, J.M. Study of the last part of the stress-deformation curve of construction steels with distinct fracture patterns. Engineering Fracture Mechanics 2016, 166, 43 – 59. doi: http://dx.doi.org/10.1016/j.engfracmech.2016.08.022. - Suárez, F.; Gálvez, J.C.; Cendón, D.A.; Atienza, J.M. Fracture of eutectoid steel bars under tensile loading: Experimental results and numerical simulation. Engineering Fracture Mechanics 2016, 158, 87 – 105. doi: http://dx.doi.org/10.1016/j.engfracmech.2016.02.044.

            To provide a more updated state of the art, the last paragraph of the introduction has been modified including new references.       

            In regards to the similar figures used in previous papers, which has also been pointed out by Reviewer 1, several modifications have been done:

       Regarding Figures 5 and 6: we have removed the general views of the SEM analysis of both materials, leaving only the fractographs that allow identifying distinct fracture mechanisms, which are commented later in the paper.

       Regarding Figures 8, 9 and 10: we have included a sentence at the beginning of section 2.4.3, where the reader is warned that some of these results in these figures were already included in one of the references, but are now completed with more images to better understand the damage process in each material.

3- For the numerical model is necessary to describe how it is considered the porosity in the material characterization. How could affect the porosity in the mechanical properties?.

            Material porosity is introduced by parameter d. When model parameters are presented in section 3.2 there is a note clarifying that, since the Abaqus implementation of the model is used, porosity is specified by this parameter, which represents “density”. Therefore, a value of 1 means no porostiy at all and a value of 0.9 means a porosity of 10%.

4- There is a lack of discussion of the presented results with other research papers to compared the results and state what novelty is bringing this research work and how can be used in future for other researcher.

            The discussion of results has been modified and extended. The last section, now called “Conclusions and final remarks” has been modified and now highlights more clearly one of the main ideas extracted from this work, which can help researchers to study and improve the calibration of the models based on the formulation proposed by Gurson.

5- The conclusion section is coarse and some paragraphs, like the last three, are discussion instead of conclusions bullets. It is necessary to explicitly described the originality of this work with respect to other research works presented in the same research topic and how can it help this research to the industry.

            This section has been renamed as “Conclusions and final remarks” and a final paragraph has been included to highlight one of the main ideas.

Round  2

Reviewer 2 Report

The authors carefully revised the manuscript taking into account the comments and suggestions. I only point out a few minor comment and suggestions.

COMMENT: In page 5, the authors’ state “For each of them, a general fractograph of the fracture surface and two closer fractographs are shown, one of the central region of the surface and the other of the external region. The internal region is highlighted by a dotted circumference. In the Material 1 specimen it corresponds to the internal dark region observed after a tensile test and in the Material 2 it corresponds to the flat surface perpendicular to the specimen axis in a cup-cone fracture surface.”

Unfortunately, after the changes the reference to the “dotted circumference” is no longer valid. Thus, the authors must either revise the text or include the old version of the figures.

SUGESTION: In page 1, replace “until,at a” by “until, at a”

SUGESTION: In page 2, replace “Subsequent evolution” by “Subsequent evolutions”

SUGESTION: In page 3, replace “numerically obtained” by “numerically predicted”

SUGESTION: In page 6, replace “Please,note” by ““Please, note”

SUGESTION: In page 14, remove the paragraph in line 317.

Author Response

Dear Editor,

Please, find enclosed the updated revised version of the following paper:

Journal:                      Metals

Manuscript ID:           metals-419571

Title:                          The evolution of internal damage identified by means of X-ray computed                                   tomography in two steels and the ensuing relation with Gurson’s numerical                          modelling

Author:                       F. Suárez, F. Sket, J.C. Gálvez, D.A. Cendón, J.M. Atienza, J. Molina-Aldareguia

This revision responds the last reviewer, (Reviewer 1) whose revision was received by us on the 23rdof January, after we sent a former revision responding the other three reviewers. We would like to thank this reviewer for his comments, although we had already addressed several changes thanks to the other reviewers, this has helped us correct some issues and improve the manuscript.

We also respond the second round of Reviewer 2, which has helped us polish the final version, especially with his first comment.

Below we address each of the remarks made by the authors (in blue) to the comments of Reviewer 1 and reference the changes made in the manuscript (highlighted in green in the new version). Note that there are more modifications that correspond to other reviewers’ suggestions, which are highlighted in yellow. Finally, we also include the modifications suggested by Reviewer 2, which are highlighted in blue in the new version.

Yours faithfully,

The authors

Comments to reviewers

Reviewer #2 (round 2)

The authors carefully revised the manuscript taking into account the comments and suggestions. I only point out a few minor comment and suggestions.

COMMENT: In page 5, the authors’ state “For each of them, a general fractograph of the fracture surface and two closer fractographs are shown, one of the central region of the surface and the other of the external region. The internal region is highlighted by a dotted circumference. In the Material 1 specimen it corresponds to the internal dark region observed after a tensile test and in the Material 2 it corresponds to the flat surface perpendicular to the specimen axis in a cup-cone fracture surface.”

Unfortunately, after the changes the reference to the “dotted circumference” is no longer valid. Thus, the authors must either revise the text or include the old version of the figures.

            We have kept the new version of the figures and have revised the text and corrected it accordingly.

SUGESTION: In page 1, replace “until,at a” by “until, at a”

SUGESTION: In page 2, replace “Subsequent evolution” by “Subsequent evolutions”

SUGESTION: In page 3, replace “numerically obtained” by “numerically predicted”

SUGESTION: In page 6, replace “Please,note” by ““Please, note”

SUGESTION: In page 14, remove the paragraph in line 317.

            All these suggested changes have been applied in the last version of the manuscript. We wish to thank the reviewer for his very careful revision of the text.

Reviewer 4 Report

 The authors have strengthen the manuscript, so in my opinion can be accepted in its present from.

Author Response

The authors thank the kind comment of the reviewer.